# DIFFUSION MODEL FOR DENSE MATCHING

**Jisu Nam**[1]  **Gyuseong Lee**[2*]  **Sunwoo Kim**[3*]  **Hyeonsu Kim**[1]  **Hyoungwon Cho**[1]
**Seyeon Kim**[1]  **Seungryong Kim**[1†]
[1]Korea University  [2]LG Electronics  [3]KT

## ABSTRACT

The objective for establishing dense correspondence between paired images consists of two terms: a data term and a prior term. While conventional techniques focused on defining hand-designed prior terms, which are difficult to formulate, recent approaches have focused on learning the data term with deep neural networks without explicitly modeling the prior, assuming that the model itself has the capacity to learn an optimal prior from a large-scale dataset. The performance improvement was obvious, however, they often fail to address inherent ambiguities of matching, such as textureless regions, repetitive patterns, large displacements, or noises. To address this, we propose DiffMatch, a novel conditional diffusion-based framework designed to explicitly model both the data and prior terms for dense matching. This is accomplished by leveraging a conditional denoising diffusion model that explicitly takes matching cost and injects the prior within generative process. However, limited input resolution of the diffusion model is a major hindrance. We address this with a cascaded pipeline, starting with a low-resolution model, followed by a super-resolution model that successively upsamples and incorporates finer details to the matching field. Our experimental results demonstrate significant performance improvements of our method over existing approaches, and the ablation studies validate our design choices along with the effectiveness of each component. Code and pretrained weights are available at https://ku-cvlab.github.io/DiffMatch.

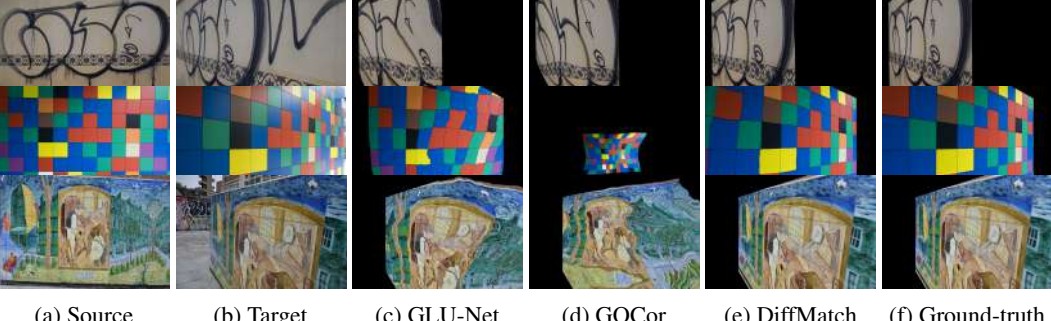

|  (a) Source | (b) Target | (c) GLU-Net | (d) GOCor | (e) DiffMatch | (f) Ground-truth |

Figure 1: **Visualizing the effectiveness of the proposed DiffMatch:** (a) source images, (b) target images, and warped source images using estimated correspondences by (c-d) state-of-the-art approaches (Truong et al., 2020b;a), (e) our DiffMatch, and (f) ground-truth. Compared to previous methods (Truong et al., 2020b;a) that *discriminatively* estimate correspondences, our diffusion-based *generative* framework effectively learns the matching field manifold, resulting in better estimating correspondences particularly at textureless regions, repetitive patterns, and large displacements.

## 1 INTRODUCTION

Establishing pixel-wise correspondences between pairs of images has been one of the crucial problems, as it supports a wide range of applications, including structure from motion (SfM) (Schonberger & Frahm, 2016), simultaneous localization and mapping (SLAM) (Durrant-Whyte & Bailey, 2006;

---

*Work done while at Korea University.
†Corresponding author.

Bailey & Durrant-Whyte, 2006), image editing (Barnes et al., 2009; Cheng et al., 2010; Zhang et al., 2020), and video analysis (Hu et al., 2018; Lai & Xie, 2019). In contrast to sparse correspondence (Calonder et al., 2010; Lowe, 2004; Sarlin et al., 2020) which detects and matches only a sparse set of key points, dense correspondence (Pérez et al., 2013; Rocco et al., 2017; Kim et al., 2018; Cho et al., 2021) aims to match all the points between input images.

In the probabilistic interpretation, the objective for dense correspondence can be defined with a *data* term, measuring matching evidence between source and target features, and a *prior* term, encoding prior knowledge of correspondence. Traditional methods (Pérez et al., 2013; Drulea & Nedevschi, 2011; Werlberger et al., 2010; Lhuillier & Quan, 2000; Liu et al., 2010; Ham et al., 2016) explicitly incorporated hand-designed prior terms to achieve smoother correspondence, such as total variation (TV) or image discontinuity-aware smoothness. However, the formulation of the hand-crafted prior term is notoriously challenging and may vary depending on the specific dense correspondence tasks, such as geometric matching (Liu et al., 2010; Duchenne et al., 2011; Kim et al., 2013) or optical flow (Weinzaepfel et al., 2013; Revaud et al., 2015).

Unlike them, recent approaches (Kim et al., 2017a; Sun et al., 2018; Rocco et al., 2017; 2020; Truong et al., 2020b; Min & Cho, 2021; Kim et al., 2018; Jiang et al., 2021; Cho et al., 2021; 2022) have focused on solely learning the data term with deep neural networks. However, despite demonstrating certain performance improvements, these methods still struggle with effectively addressing the inherent ambiguities encountered in dense correspondence, including challenges posed by textureless regions, repetitive patterns, large displacements, or noises. We argue that it is because they concentrate on maximizing the likelihood, which corresponds to learning the data term only, and do not *explicitly* consider the matching prior. This limits their ability to learn ideal matching field manifold, and leads to poor generalization.

On the other hand, diffusion models (Ho et al., 2020; Song et al., 2020a; Song & Ermon, 2019; Song et al., 2020b) have recently demonstrated a powerful capability for learning posterior distribution and have achieved considerable success in the field of generative models (Karras et al., 2020). Building on these advancements, recent studies (Rombach et al., 2022; Seo et al., 2023; Saharia et al., 2022a; Lugmayr et al., 2022) have focused on controllable image synthesis by leveraging external conditions. Moreover, these advances in diffusion models have also led to successful applications in numerous discriminative tasks, such as depth estimation (Saxena et al., 2023b; Kim et al., 2022; Duan et al., 2023), object detection (Chen et al., 2022a), segmentation (Gu et al., 2022; Giannone et al., 2022), and human pose estimation (Holmquist & Wandt, 2022).

Inspired by the recent success of the diffusion model (Ho et al., 2020; Song et al., 2020a; Song & Ermon, 2019; Song et al., 2020b), we introduce DiffMatch, a conditional diffusion-based framework designed to *explicitly* model the matching field distribution within diffusion process.

Unlike existing discriminative learning-based methods (Kim et al., 2017a; Jiang et al., 2021; Rocco et al., 2017; 2020; Teed & Deng, 2020) that focus solely on maximizing the *likelihood*, DiffMatch aims to learn the *posterior* distribution of dense correspondence. Specifically, this is achieved by a conditional denoising diffusion model designed to learn how to generate a correspondence field given feature descriptors as conditions. However, limited input resolution of the diffusion model is a significant hindrance. To address this, we adopt a cascaded diffusion pipeline, starting with a low-resolution diffusion model, and then transitioning to a super-resolution diffusion model that successively upsamples the matching field and incorporates higher-resolution details.

We evaluate the effectiveness of DiffMatch using several standard benchmarks (Balntas et al., 2017; Schops et al., 2017), and show the robustness of our model with the corrupted datasets (Hendrycks & Dietterich, 2019; Balntas et al., 2017; Schops et al., 2017). We also conduct extensive ablation studies to validate our design choices and explore the effectiveness of each component.

## 2 RELATED WORK

**Dense correspondence.** Traditional methods for dense correspondence (Horn & Schunck, 1981; Lucas & Kanade, 1981) relied on hand-designed matching priors. Several techniques (Sun et al., 2010; Brox & Malik, 2010; Liu et al., 2010; Taniai et al., 2016; Kim et al., 2017b; Ham et al., 2016; Kim et al., 2013) introduced optimization methods, such as SIFT Flow (Liu et al., 2010), which designed smoothness and small displacement priors, and DCTM (Kim et al., 2017b), which

introduced a discontinuity-aware prior term. However, manually designing the prior term is difficult. To address this, recent approaches (Dosovitskiy et al., 2015; Rocco et al., 2017; Shen et al., 2019; Melekhov et al., 2019; Ranjan & Black, 2017; Teed & Deng, 2020; Sun et al., 2018; Truong et al., 2020b; 2021; Jiang et al., 2021) have shifted to a learning paradigm, formulating an objective function to solely maximize likelihood. This assumes that an optimal matching prior can be learned from a large-scale dataset. DGC-Net (Melekhov et al., 2019) and GLU-Net (Truong et al., 2020b) proposed a coarse-to-fine framework using a feature pyramid, while COTR (Jiang et al., 2021) employed a transformer-based network. GOCor (Truong et al., 2020a) developed a differentiable matching module to learn spatial priors, addressing matching ambiguities. PDC-Net+(Truong et al., 2023) presented dense matching using a probabilistic model, estimating a flow field paired with a confidence map. DKM (Edstedt et al., 2023) introduced a kernel regression global matcher to find accurate global matches and their certainty.

**Diffusion models.** Diffusion models (Sohl-Dickstein et al., 2015; Ho et al., 2020) have been extensively researched due to their powerful generation capability. The Denoising Diffusion Probabilistic Models (DDPM) (Ho et al., 2020) proposed a diffusion model in which the forward and reverse processes exhibit the Markovian property. The Denoising Diffusion Implicit Models (DDIM) (Song et al., 2020a) accelerated DDPM by replacing the original diffusion process with non-Markovian chains to enhance the sampling speed. Building upon these advancements, conditional diffusion models that leverage auxiliary conditions for controlled image synthesis have emerged. Palette (Saharia et al., 2022a) proposed a general framework for image-to-image translation by concatenating the source image as an additional condition. Similarly, InstructPix2Pix (Brooks et al., 2023) trains a conditional diffusion model using a paired image and text instruction, specifically tailored for instruction-based image editing. On the other hand, several studies (Ho et al., 2022; Saharia et al., 2022b; Ryu & Ye, 2022; Balaji et al., 2022) have turned their attention to resolution enhancement, as the Cascaded Diffusion Model (Ho et al., 2022) adopts a cascaded pipeline to progressively interpolate the resolution of synthesized images using the diffusion denoising process.

**Diffusion model for discriminative tasks.** Recently, the remarkable performance of the diffusion model has been extended to solve discriminative tasks, including image segmentation (Chen et al., 2022b; Gu et al., 2022; Ji et al., 2023), depth estimation (Saxena et al., 2023b; Kim et al., 2022; Duan et al., 2023; Ji et al., 2023), object detection (Chen et al., 2022a), and pose estimation (Tevet et al., 2022; Holmquist & Wandt, 2022). These approaches have demonstrated noticeable performance improvement using diffusion models. Our method represents the first application of the diffusion model to the dense correspondence task.

## 3  PRELIMINARIES

**Probabilistic interpretation of dense correspondence.** Let us denote a pair of images, i.e., source and target, as $I_{\mathrm{src}}$ and $I_{\mathrm{tgt}}$ that represent visually or semantically similar images, and feature descriptors extracted from $I_{\mathrm{src}}$ and $I_{\mathrm{tgt}}$ as $D_{\mathrm{src}}$ and $D_{\mathrm{tgt}}$, respectively. The objective of dense correspondence is to find a correspondence field $F$ that is defined at each pixel $i$, which warps $I_{\mathrm{src}}$ towards $I_{\mathrm{tgt}}$ such that $I_{\mathrm{tgt}}(i) \sim I_{\mathrm{src}}(i + F(i))$ or $D_{\mathrm{tgt}}(i) \sim D_{\mathrm{src}}(i + F(i))$.

This objective can be formulated within probabilistic interpretation (Simoncelli et al., 1991; Sun et al., 2008; Ham et al., 2016; Kim et al., 2017b), where we seek to find $F^*$ that maximizes the posterior probability of the correspondence field given a pair of feature descriptors $D_{\mathrm{src}}$ and $D_{\mathrm{tgt}}$, i.e., $p(F|D_{\mathrm{src}}, D_{\mathrm{tgt}})$. According to Bayes' theorem (Joyce, 2003), the posterior can be decomposed such that $p(F|D_{\mathrm{src}}, D_{\mathrm{tgt}}) \propto p(D_{\mathrm{src}}, D_{\mathrm{tgt}}|F) \cdot p(F)$. To find the matching field $F^*$ that maximizes the posterior, we can use the maximum a posteriori (MAP) approach (Greig et al., 1989):

$$
\begin{aligned}
F^* &= \operatorname*{argmax}_{F} p(F|D_{\mathrm{src}}, D_{\mathrm{tgt}}) = \operatorname*{argmax}_{F} p(D_{\mathrm{src}}, D_{\mathrm{tgt}}|F) \cdot p(F) \\
&= \operatorname*{argmax}_{F} \{ \underbrace{\log p(D_{\mathrm{src}}, D_{\mathrm{tgt}}|F)}_{\text{data term}} + \underbrace{\log p(F)}_{\text{prior term}} \}.
\end{aligned}
\tag{1}
$$

In this probabilistic interpretation, the first term, referred to as *data* term, represents the matching evidence between feature descriptors $D_{\mathrm{src}}$ and $D_{\mathrm{tgt}}$, and the second term, referred to as *prior* term, encodes prior knowledge of the matching field $F$.

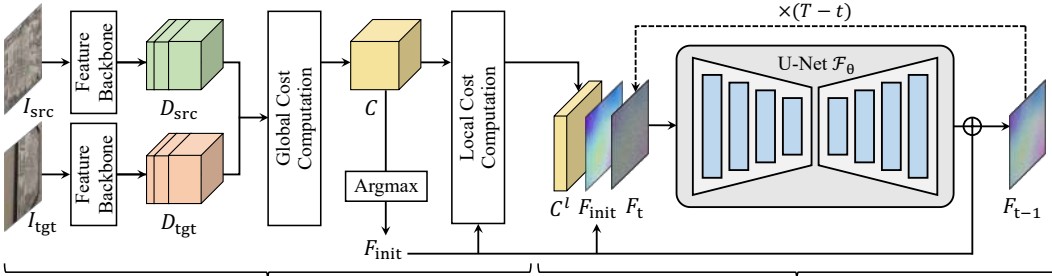

Figure 2: **Overall network architecture of DiffMatch.** Given source and target images, our conditional diffusion-based network estimates the dense correspondence between the two images. We leverage two conditions: the initial correspondence $F_{\text{init}}$ and the local matching cost $C^l$, which finds long-range matching and embeds local pixel-wise interactions, respectively.

**Conditional diffusion models.** The diffusion model is a type of generative model, and can be divided into two categories: unconditional models (Sohl-Dickstein et al., 2015; Ho et al., 2020) and conditional models (Batzolis et al., 2021; Dhariwal & Nichol, 2021). Specifically, unconditional diffusion models learn an explicit approximation of the data distribution, denoted as $p(X)$. On the other hand, conditional diffusion models estimate the data distribution given a certain condition $K$, e.g., text prompt (Dhariwal & Nichol, 2021), denoted as $p(X|K)$.

In the conditional diffusion model, the data distribution is approximated by recovering a data sample from the Gaussian noise through an iterative denoising process. Given a sample $X_0$, it is transformed to $X_t$ through the forward diffusion process at a time step $t \in \{T, T - 1, \ldots, 1\}$, which consists of Gaussian transition at each time step $q(X_t|X_{t-1}) := \mathcal{N}(\sqrt{1 - \beta_t}X_{t-1}, \beta_t I)$. The forward diffusion process follows the pre-defined variance schedule $\beta_t$ such that

$$X_t = \sqrt{\alpha_t}X_0 + \sqrt{1 - \alpha_t}Z, \quad Z \sim \mathcal{N}(0, I), \tag{2}$$

where $\alpha_t = \prod_{i=1}^{t}(1 - \beta_i)$. After training, we can sample data from the learned distribution through iterative denoising with the pre-defined range of time steps, called the reverse diffusion process, following the non-Markovian process of DDIM (Song et al., 2020a), which is parametrized as another Gaussian transition $p_\theta(X_{t-1} \mid X_t) := \mathcal{N}(X_{t-1}; \mu_\theta(X_t, t), \sigma_\theta(X_t, t)I)$. To this end, the diffusion network $\mathcal{F}_\theta(X_t, t; K)$ predicts the denoised sample $\hat{X}_{0,t}$ given $X_t$, $t$ and $K$. One step in the reverse diffusion process can be formulated such that

$$X_{t-1} = \sqrt{\alpha_{t-1}}\mathcal{F}_\theta(X_t, t; K) + \frac{\sqrt{1 - \alpha_{t-1} - \sigma_t^2}}{\sqrt{1 - \alpha_t}}\left(X_t - \sqrt{\alpha_t}\mathcal{F}_\theta(X_t, t; K)\right) + \sigma_t Z \tag{3}$$

where $\sigma_t$ is the covariance value of Gaussian distribution at time step $t$.

This iterative denoising process can be viewed as finding $X^* = \operatorname{argmax}_X \log p(X|K)$ through the relationship between the conditional sampling process of DDIM (Song et al., 2020a) and conditional score-based generative models (Batzolis et al., 2021).

## 4 METHODOLOGY

### 4.1 MOTIVATION

Recent learning-based methods (Kim et al., 2017a; Sun et al., 2018; Rocco et al., 2017; 2020; Truong et al., 2020b; Min & Cho, 2021; Kim et al., 2018; Jiang et al., 2021; Cho et al., 2021; 2022) have employed deep neural networks $\mathcal{F}(\cdot)$ to directly approximate the *data* term, i.e., $\operatorname{argmax}_F \log p(D_{\text{src}}, D_{\text{tgt}}|F)$, without explicitly considering the *prior* term. For instance, GLU-Net (Truong et al., 2020b) and GOCor (Truong et al., 2020a) construct a cost volume along candidates $F$ between source and target features $D_{\text{src}}$ and $D_{\text{tgt}}$, and regresses the matching fields $F^*$ within deep neural networks, which might be analogy to $\operatorname{argmax}_F \log p(D_{\text{src}}, D_{\text{tgt}}|F)$. In this setting,

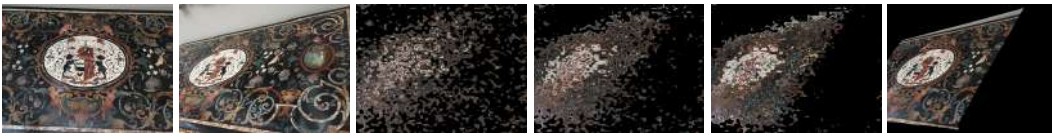

Figure 3: **Visualization of the reverse diffusion process in DiffMatch:** (from left to right) source and target images, and warped source images by estimated correspondences as evolving time steps. The source image is progressively warped into the target image through an iterative denoising process.

dense correspondence $F^*$ is estimated as follows:

$$F^* = \mathcal{F}_\theta(D_{\mathrm{src}}, D_{\mathrm{tgt}}) \approx \underset{F}{\mathrm{argmax}} \underbrace{\log p(D_{\mathrm{src}}, D_{\mathrm{tgt}}|F)}_{\text{data term}}, \tag{4}$$

where $\mathcal{F}_\theta(\cdot)$ and $\theta$ represent a feed-forward network and its parameters, respectively.

These approaches assume that the matching prior can be learned within the model architecture by leveraging the high capacity of deep networks (Truong et al., 2020b; Jiang et al., 2021; Cho et al., 2021; 2022; Min & Cho, 2021) and the availability of large-scale datasets. While there exists obvious performance improvement, they typically focus on the data term without explicitly considering the matching prior. This can restrict ability of the model to learn the manifold of matching field and result in poor generalization.

## 4.2 FORMULATION

To address these limitations, for the first time, we explore a conditional generative model for dense correspondence to explicitly learn both the *data* and *prior* terms. Unlike previous discriminative learning-based approaches (Pérez et al., 2013; Drulea & Nedevschi, 2011; Werlberger et al., 2010; Kim et al., 2017a; Sun et al., 2018; Rocco et al., 2017), we achieve this by leveraging a conditional generative model that jointly learns the data and prior through optimization of the following objective that *explicitly* learn $\mathrm{argmax}_F p(F|D_{\mathrm{src}}, D_{\mathrm{tgt}})$:

$$\begin{aligned} F^* = \mathcal{F}_\theta(D_{\mathrm{src}}, D_{\mathrm{tgt}}) &\approx \underset{F}{\mathrm{argmax}}\, p(F|D_{\mathrm{src}}, D_{\mathrm{tgt}}) \\ &= \underset{F}{\mathrm{argmax}} \{ \underbrace{\log p(D_{\mathrm{src}}, D_{\mathrm{tgt}}|F)}_{\text{data term}} + \underbrace{\log p(F)}_{\text{prior term}} \}. \end{aligned} \tag{5}$$

We leverage the capacity of a conditional diffusion model, which generates high-fidelity and diverse samples aligned with the given conditions, to search for accurate matching within the learned correspondence manifold.

Specifically, we define the forward diffusion process for dense correspondence as the Gaussian transition such that $q(F_t|F_{t-1}) := \mathcal{N}(\sqrt{1-\beta_t}F_{t-1}, \beta_t I)$, where $\beta_t$ is a predefined variance schedule. The resulting latent variable $F_t$ can be formulated as Eq. 2:

$$F_t = \sqrt{\alpha_t}F_0 + \sqrt{1-\alpha_t}Z, \quad Z \sim \mathcal{N}(0, I), \tag{6}$$

where $F_0$ is the ground-truth correspondence. In addition, the neural network $\mathcal{F}_\theta(\cdot)$ is subsequently trained to reverse the forward diffusion process. During the reverse diffusion phase, the initial latent variable $F_T$ is iteratively denoised following the sequence $F_{T-1}, F_{T-2}, \ldots, F_0$, using Eq. 3:

$$F_{t-1} = \sqrt{\alpha_{t-1}}\mathcal{F}_\theta(X_t, t; D_{\mathrm{src}}, D_{\mathrm{tgt}}) + \frac{\sqrt{1-\alpha_{t-1}-\sigma_t^2}}{\sqrt{1-\alpha_t}}\left(X_t - \sqrt{\alpha_t}\mathcal{F}_\theta(F_t, t; D_{\mathrm{src}}, D_{\mathrm{tgt}})\right) + \sigma_t Z, \tag{7}$$

where $\mathcal{F}_\theta(F_t, t; D_{\mathrm{src}}, D_{\mathrm{tgt}})$ directly predicts the denoised correspondence $\hat{F}_{0,t}$ with source and target features, $D_{\mathrm{src}}$ and $D_{\mathrm{tgt}}$, as conditions.

The objective of this denoising process is to find the optimal correspondence field $F^*$ that satisfies $\mathrm{argmax}_F \log p(F|D_{\mathrm{src}}, D_{\mathrm{tgt}})$. The detailed explanation of the objective function of the denoising process will be explained in Section 4.5.

## 4.3 Network architecture

In this section, we discuss how to design the network architecture $\mathcal{F}_\theta(\cdot)$. Our goal is to find accurate matching fields given feature descriptors $D_\mathrm{src}$ and $D_\mathrm{tgt}$ from $I_\mathrm{src}$ and $I_\mathrm{tgt}$, respectively, as conditions. An overview of our proposed architecture is provided in Figure 2.

**Cost computation.** Following conventional methods (Rocco et al., 2020; Truong et al., 2020a), we first compute the matching cost by calculating the pairwise cosine similarity between localized deep features from the source and target images. Given image features $D_\mathrm{src}$ and $D_\mathrm{tgt}$, the matching cost is constructed by taking scalar products between all locations in the feature descriptors, formulated as:

$$C(i,j) = \frac{D_\mathrm{src}(i) \cdot D_\mathrm{tgt}(j)}{\|D_\mathrm{src}(i)\| \|D_\mathrm{tgt}(j)\|}, \tag{8}$$

where $i \in [0, h_\mathrm{src}) \times [0, w_\mathrm{src})$, $j \in [0, h_\mathrm{tgt}) \times [0, w_\mathrm{tgt})$, and $\|\cdot\|$ denotes $l$-2 normalization.

Forming the global matching cost by computing all pairwise feature dot products is robust to long-range matching. However, it is computationally unfeasible due to its high dimensionality such that $C \in \mathbb{R}^{h_\mathrm{src} \times w_\mathrm{src} \times h_\mathrm{tgt} \times w_\mathrm{tgt}}$. To alleviate this, we can build the local matching cost by narrowing down the target search region $j$ within a neighborhood of the source location $i$, constrained by a search radius $R$. Compared to the global matching cost, the local matching cost $C^l \in \mathbb{R}^{h_\mathrm{src} \times w_\mathrm{src} \times R \times R}$ is suitable for small displacements and, thanks to its constrained search range of $R$, is more feasible for large spatial sizes and can be directly used as a condition for the diffusion model. Importantly, the computational overhead remains minimal, with the only significant increase being $R \times R$ in the channel dimension.

**Conditional denoising diffusion model.** As illustrated in Figure 2, we introduce a modified U-Net architecture based on (Nichol & Dhariwal, 2021). Our aim is to generate an accurate matching field that aligns with the given conditions. A direct method to condition the model is simply concatenating $D_\mathrm{src}$ and $D_\mathrm{tgt}$ with the noisy flow input $F_t$. However, this led to suboptimal performance in our tests. Instead, we present two distinct conditions for our network: the initial correspondence and the local matching cost.

First, our model is designed to learn the residual of the initially estimated correspondence, which leads to improved initialization and enhanced stability. Specifically, we calculate the initial correspondence $F_\mathrm{init}$ using the soft-argmax operation (Cho et al., 2021) based on the global matching cost $C$ between $D_\mathrm{src}$ and $D_\mathrm{tgt}$. This assists the model to find long-range matches. Secondly, we integrate pixel-wise interactions between paired images. For this, each pixel $i$ in the source image is mapped to $i'$ in the target image through the estimated initial correspondence $F_\mathrm{init}$. We then compute the local matching cost $C^l$ as an additional condition with $F_\mathrm{init}$. This local cost guides the model to focus on the neighborhood of the initial estimation, helping to find a more refined matching field. With these combined, our conditioning strategies enable the model to precisely navigate the matching field manifold while preserving its generative capability. Finally, under the conditions $F_\mathrm{init}$ and $C^l$, the noised matching field $F_t$ at time step $t$ passes through the modified U-net (Nichol & Dhariwal, 2021), which comprises convolution and attention, and generates the denoised matching field $\hat{F}_{t,0}$ aligned with the given conditions.

## 4.4 Flow upsampling

The inherent input resolution limitations of the diffusion model is a major hindrance. Inspired by recent super-resolution diffusion models (Ho et al., 2022; Ryu & Ye, 2022), we propose a cascaded pipeline tailored for flow upsampling. Our approach begins with a low-resolution denoising diffusion model, followed by a super-resolution model, successively upsampling and adding fine-grained details to the matching field. To achieve this, we simply finetune the pre-trained conditional denoising diffusion model, which was trained at a coarse resolution. Specifically, instead of using $F_\mathrm{init}$ from the global matching cost, we opt for a downsampled ground-truth flow field as $F_\mathrm{init}$. This simple modification effectively harnesses the power of the pretrained diffusion model for flow upsampling. The efficacy of our flow upsampling model is demonstrated in Table 4.

Table 1: **Quantitative evaluation on HPatches (Balntas et al., 2017) and ETH3D (Schops et al., 2017) with common corruptions from ImageNet-C (Hendrycks & Dietterich, 2019).** All results are evaluated at corruption severity 5. For simplicity, we denote GLU-Net-GOCor as GOCor (Truong et al., 2020a).

| Dataset | Algorithm | Noise | | | Blur | | | | Weather | | | | Digital | | | | Avg. |
|---|---|---|---|---|---|---|---|---|---|---|---|---|---|---|---|---|---|
| | | Gauss. | Shot | Impulse | Defocus | Glass | Motion | Zoom | Snow | Frost | Fog | Bright | Contrast | Elastic | Pixel | JPEG | |
| HPatches | GLU-Net (Truong et al., 2020b) | 34.96 | 32.60 | 34.18 | 25.74 | 25.71 | 63.26 | 90.75 | 46.16 | 66.63 | 47.81 | 25.28 | 37.45 | 32.85 | 44.31 | 26.94 | 42.31 |
| | GOCor (Truong et al., 2020a) | **27.35** | **27.21** | **26.63** | 23.54 | 20.75 | 57.75 | 88.35 | 39.84 | 63.55 | 36.98 | 21.44 | 23.65 | 28.40 | _33.67_ | _22.20_ | 36.09 |
| | PDCNet (Truong et al., 2021) | 30.00 | 29.97 | 29.36 | 25.94 | 24.06 | 56.96 | _85.44_ | 42.31 | _56.87_ | 40.98 | 23.16 | _23.29_ | 29.52 | 34.10 | 23.55 | 37.03 |
| | PDCNet+ (Truong et al., 2023) | _29.82_ | _27.23_ | 28.31 | _21.97_ | **19.15** | _48.29_ | 81.73 | **35.00** | 82.84 | 35.34 | 17.85 | 21.90 | 27.19 | 33.00 | 22.70 | _35.49_ |
| | **DiffMatch** | 31.10 | 28.21 | 29.14 | **21.96** | _19.56_ | **38.16** | 97.22 | _37.49_ | **50.74** | _35.66_ | **20.21** | 27.22 | _27.43_ | 37.17 | **21.63** | **34.86** |
| ETH3D | GLU-Net (Truong et al., 2020b) | 29.20 | 27.51 | 29.11 | 14.18 | 13.16 | 36.90 | 77.73 | 47.11 | _65.22_ | 37.75 | 13.89 | 24.52 | 19.11 | 21.25 | 17.77 | 31.63 |
| | GOCor (Truong et al., 2020a) | _27.45_ | 25.56 | _26.44_ | 11.39 | 10.98 | _33.73_ | 73.56 | 43.24 | 66.49 | 39.11 | 10.98 | 18.34 | 15.54 | _16.47_ | 15.20 | 28.96 |
| | PDCNet (Truong et al., 2021) | 28.60 | _24.94_ | 27.90 | 10.63 | 10.00 | 37.93 | 76.18 | 45.08 | 69.50 | 35.90 | 10.16 | 17.46 | 15.86 | 16.69 | 15.62 | 29.50 |
| | PDCNet+ (Truong et al., 2023) | 30.49 | 26.18 | 28.08 | _8.99_ | _8.32_ | 33.40 | 68.79 | 39.14 | 65.35 | _31.50_ | _8.59_ | 8.59 | _14.55_ | 14.55 | 13.28 | _27.11_ |
| | **DiffMatch** | 25.11 | 23.36 | 24.61 | **8.62** | **5.48** | 36.47 | _72.67_ | _41.48_ | 64.82 | 25.68 | 8.13 | _15.32_ | 12.86 | 17.32 | _14.86_ | 26.45 |

(a) Source   (b) Target   (c) GLU-Net   (d) GOCor   (e) PDCNet+   (f) DiffMatch   (g) GT

Figure 4: **Qualitative results on HPatches (Balntas et al., 2017) using motion blur in Hendrycks & Dietterich (2019).** The source images are warped to the target images using predicted correspondences.

## 4.5 TRAINING

In training phase, the denoising diffusion model, as illustrated in Section 4.3, learns the prior knowledge of the matching field with the initial correspondence $F_{\text{init}}$ to give a matching hint and the local matching cost $C^l$ to provide additional pixel-wise interactions. In other words, we redefine the network $\mathcal{F}_\theta(F_t, t; D_{\text{src}}, D_{\text{tgt}})$ as $\mathcal{F}_\theta(F_t, t; F_{\text{init}}, C^l)$, given that $F_{\text{init}}$ and $C^l$ are derived from $D_{\text{src}}$ and $D_{\text{tgt}}$ as described in Section 4.3. The loss function for training diffusion model is defined as follows:

$$\mathcal{L} = \mathbb{E}_{F_0, t, Z \sim \mathcal{N}(0, I), D_{\text{src}}, D_{\text{tgt}}} \left[ \left\| F_0 - \mathcal{F}_\theta(F_t, t; F_{\text{init}}, C^l) \right\|^2 \right]. \tag{9}$$

Note that for the flow upsampling diffusion model, we finetune the pretrained conditional denoising diffusion model with the downsampled ground-truth flow as $F_{\text{init}}$.

## 4.6 INFERENCE

During the inference phase, a Gaussian noise $F_T$ is gradually denoised into a more accurate matching field $F_0$ under the given features $D_{\text{src}}$ and $D_{\text{tgt}}$ as conditions through the diffusion reverse process. To account for the stochastic nature of diffusion-based models, we propose utilizing multiple hypotheses by computing the mean of the estimated multiple matching fields from multiple initializations $F_T$, which helps to reduce stochasticity of model while improving the matching performance. Further details and analyses are available in Appendix C.2.

## 5 EXPERIMENTS

## 5.1 IMPLEMENTATION DETAILS

For the feature extractor backbone, we used VGG-16 (Chatfield et al., 2014) and kept all parameters frozen throughout all experiments. Our diffusion network is based on (Nichol & Dhariwal, 2021) with modifications to the channel dimension. The network was implemented using PyTorch (Paszke et al., 2019) and trained with the AdamW optimizer (Loshchilov & Hutter, 2017) at a learning rate of 1e−4 for the denoising diffusion model and 3e−5 for flow upsampling model. We conducted comprehensive experiments in geometric matching for four datasets: HPatches (Balntas et al., 2017), ETH3D (Schops et al., 2017), ImageNet-C (Hendrycks & Dietterich, 2019) corrupted HPatches and ImageNet-C corrupted ETH3D. Following (Truong et al., 2020b;a; 2023), we trained our network using DPED-CityScape-ADE (Ignatov et al., 2017; Cordts et al., 2016; Zhou et al., 2019) and COCO (Lin et al., 2014)-augmented DPED-CityScape-ADE for evaluation on Hpatches and ETH3D, respectively. For a fair comparison, we benchmark our method against PDCNet (Truong et al., 2021)

Table 2: **Quantitative evaluation on HPatches (Balntas et al., 2017) and ETH3D (Schops et al., 2017).** Lower AEPE indicates better performance. Higher scene labels or rates (e.g., V or 15) comprise more challenging images with extreme geometric deformations. The best results are highlighted in bold, and the second-best results are underlined. *: COTR (Jiang et al., 2021) is examined separately since it provides only confident correspondences and evaluation is limited to this subset. †: This indicates that a dense evaluation is performed without zoom-in techniques and confidence thresholding for a fair comparison.

| Methods | HPatches Original (Balntas et al., 2017) AEPE ↓ | | | | | | ETH3D (Schops et al., 2017) AEPE ↓ | | | | | | | |
|---|---|---|---|---|---|---|---|---|---|---|---|---|---|---|
| | I | II | III | IV | V | Avg. | rate=3 | rate=5 | rate=7 | rate=9 | rate=11 | rate=13 | rate=15 | Avg. |
| COTR* (Jiang et al., 2021) | - | - | - | - | - | 7.75 | 1.66 | 1.82 | 1.97 | 2.13 | 2.27 | 2.41 | 2.61 | 2.12 |
| COTR*+Interp. (Jiang et al., 2021) | - | - | - | - | - | 7.98 | 1.71 | 1.92 | 2.16 | 2.47 | 2.85 | 3.23 | 3.76 | 2.59 |
| DGC-Net (Melekhov et al., 2019) | 5.71 | 20.48 | 34.15 | 43.94 | 62.01 | 33.26 | 2.49 | 3.28 | 4.18 | 5.35 | 6.78 | 9.02 | 12.25 | 6.19 |
| GLU-Net (Truong et al., 2020b) | 1.55 | 12.66 | 27.54 | 32.04 | 52.47 | 25.05 | 1.98 | 2.54 | 3.49 | 4.24 | 5.61 | 7.55 | 10.78 | 5.17 |
| GLU-Net-GOCor (Truong et al., 2020a) | **1.29** | 10.07 | 23.86 | 27.17 | 38.41 | 20.16 | 1.93 | 2.28 | 2.64 | 3.01 | 3.62 | 4.79 | 7.80 | 3.72 |
| DMP (Hong & Kim, 2021) | 3.21 | 15.54 | 32.54 | 38.62 | 63.43 | 30.64 | 2.43 | 3.31 | 4.41 | 5.56 | 6.93 | 9.55 | 14.20 | 6.62 |
| COTR† (Jiang et al., 2021) | 19.65 | 33.81 | 45.81 | 62.03 | 66.28 | 45.52 | 8.76 | 9.86 | 11.23 | 12.44 | 13.77 | 14.94 | 16.09 | 12.44 |
| PDCNet (Truong et al., 2021) | 1.30 | 11.92 | 28.60 | 35.97 | 42.41 | 24.04 | 1.77 | 2.10 | 2.50 | 2.88 | 3.47 | 4.88 | 7.57 | 3.60 |
| PDCNet+ (Truong et al., 2023) | 1.44 | **8.97** | 22.24 | 30.13 | **31.77** | **18.91** | **1.70** | **1.96** | **2.24** | **2.57** | 3.04 | 4.20 | 6.25 | 3.14 |
| **DiffMatch** | 1.85 | 10.83 | **19.18** | **26.38** | 35.96 | 18.84 | 2.08 | 2.30 | 2.59 | 2.94 | 3.29 | **3.86** | **4.54** | 3.12 |

(a) Source      (b) Target      (c) GLU-Net      (d) GOCor      (e) DiffMatch      (f) Ground-truth

Figure 5: **Qualitative results on HPatches (Balntas et al., 2017).** the source images are warped to the target images using predicted correspondences.

and PDCNet+ (Truong et al., 2023), both trained on the same synthetic dataset. Note that we strictly adhere to the training settings provided in their publicly available codebase. Further implementation details can be found in Appendix A.

## 5.2 MATCHING RESULTS

Our primary aim is to develop a robust generative prior that can effectively address inherent ambiguities in dense correspondence, such as textureless regions, repetitive patterns, large displacements, or noises. To evaluate the robustness of the proposed diffusion-based generative prior in challenging matching scenarios, we tested our approach against a series of common corruptions from ImageNet-C (Hendrycks & Dietterich, 2019). This benchmark includes 15 types of algorithmically generated corruptions, organized into four distinct categories. Additionally, we validate our method using the standard HPatches and ETH3D datasets. Further details on the corruptions and explanations for each evaluation dataset can be found in Appendix B.

**ImageNet-C corruptions.** In real-world matching scenarios, image corruptions such as weather variations or photographic distortions frequently occur. Therefore, it is crucial to establish robust dense correspondence under these corrupted conditions. However, existing discriminative methods (Truong et al., 2020b;a; 2021; 2023) solely rely on the correlation layer, focusing on point-to-point feature relationships, resulting in degraded performance in harsh-corrupted settings. In contrast, our framework learns not only the likelihood but also the prior knowledge of the matching field formation. We evaluated the robustness of our approach against the aforementioned methods on ImageNet-C corrupted scenarios (Hendrycks & Dietterich, 2019) of HPatches and ETH3D. As shown in Table 1, our method exhibits outstanding performance in harsh corruptions, especially in noise and weather. This is also visually evident in Figure 4. More qualitative results are available in Appendix D.

**HPatches.** We evaluated DiffMatch on five viewpoints of HPatches (Balntas et al., 2017). Table 2 summarizes the quantitative results and demonstrates that our method surpasses state-of-the-art discriminative learning-based methods (Truong et al., 2020b;a; 2021; 2023). The qualitative result is presented in Figure 5. The effectiveness of our approach is evident from the quantitative results in Figure 1. This success can be attributed to the robust generative prior that learns a matching field manifold, which effectively addresses challenges faced by previous discriminative methods, such as textureless regions, repetitive patterns, large displacements or noises. More qualitative results are available in Appendix D.

**ETH3D.** As indicated in Table 2, our method demonstrates highly competitive performance compared to previous discriminative works (Truong et al., 2020b;a; 2021; 2023) on ETH3D (Schops et al., 2017). Notably, DiffMatch surpasses these prior works by a large margin, especially at interval rates of 13 and 15, which represent the most challenging settings. Additional qualitative results can be found in Appendix D.

## 5.3 ABLATION STUDY

**Effectiveness of generative prior.** We aim to validate our hypothesis that a diffusion-based generative prior is effective for finding a more accurate matching field. To achieve this, we train our network by directly regressing the matching field. Then we compare its performance with our diffusion-based method. As demonstrated in Table 3, our generative approach outperforms the regression-based baseline, thereby emphasizing the efficacy of the generative prior in dense correspondence tasks. The effectiveness of the generative matching prior is further analyzed in Appendix C.3.

Table 3: **Results via different learning schemes.**

| Learning schemes | HPatches AEPE ↓ | ETH3D AEPE ↓ |
|---|---|---|
| DiffMatch w/o diffusion | 23.34 | 3.96 |
| DiffMatch | **18.82** | **3.12** |

**Component analysis.** In this ablation study, we provide a quantitative comparison between different configurations. The results are summarized in Table 4. **(I)** refers to the complete architecture of the conditional denoising diffusion model, as illustrated in Figure 2. **(II)** and **(III)** denote the conditional denoising diffusion model without the local cost and initial flow conditioning, respectively. **(IV)** represents the flow upsampling diffusion model. Notably, **(I)** outperforms both **(II)** and **(III)**, emphasizing the effectiveness of the proposed conditioning method. The comparison between **(I)** and **(IV)** underlines the benefits of the flow upsampling diffusion model, which has only a minor increase in training time as it leverages the pretrained **(I)** at a lower resolution.

Table 4: **Ablations on components.** C-ETH3D indicates ImageNet-C corrupted ETH3D.

| | Components | ETH3D AEPE ↓ | C-ETH3D AEPE ↓ |
|---|---|---|---|
| **(I)** | Conditional denoising diff. | 3.44 | 26.89 |
| **(II)** | **(I)** w/o local cost | 4.26 | 32.07 |
| **(III)** | **(I)** w/o init flow | 10.28 | 80.81 |
| **(IV)** | **(I)** + Flow upsampling diff. (DiffMatch) | **3.12** | **26.45** |

**Time complexity.** In this ablation study, we compare the time consumption of our model against existing works (Truong et al., 2021; 2023). As mentioned in Sec. 4.6, our method employs multiple hypotheses during inference, averaging them for the final output. Table 5 presents the computing times using 1, 2, and 3 samples for multiple hypotheses. Note that we employ batch processing for these multiple inputs instead of processing them sequentially, improving time efficiency. With a fixed sampling time step of 5, the time required for DiffMatch with a single input is comparable to that of previous methods (Truong et al., 2021; 2023), while ensuring comparable performance. Processing more samples leads to enhanced performance with only a negligible increase in time. Additionally, this time complexity can be further mitigated by decreasing the number of sampling time steps, as discussed in Appendix C.1.

Table 5: **Ablations on time complexity.** C-ETH3D indicates ImageNet-C corrupted ETH3D.

| Method | C-ETH3D AEPE ↓ | Time [ms] |
|---|---|---|
| PDCNet (Truong et al., 2021) | 29.50 | **112** |
| PDCNet+ (Truong et al., 2023) | 27.11 | **112** |
| DiffMatch (1 sample, 5 steps) | 27.52 | **112** |
| DiffMatch (2 samples, 5 steps) | 27.41 | 123 |
| DiffMatch (3 samples, 5 steps) | **26.45** | 140 |

## 6 CONCLUSION

In this paper, we propose DiffMatch, a novel diffusion-based framework for dense correspondence, which jointly models the likelihood and prior distribution of matching fields. This is achieved by the conditional denoising diffusion model, based on initial correspondence and local costs derived from feature descriptors. To alleviate the resolution constraint, we further propose a flow upsampling diffusion model that fine-tunes the pretrained denoising model, thereby injecting fine details into the matching field with minimal optimization. For the first time, we highlight the power of the generative prior in dense correspondence, achieving state-of-the-art performance on standard benchmarks. We further emphasize the effectiveness of our generative prior in harshly corrupted settings of the benchmarks. As a result, we demonstrate that our diffusion-based generative approach outperforms discriminative approaches in addressing the inherent ambiguities present in dense correspondence.

## ACKNOWLEDGMENTS

This research was supported by the MSIT, Korea (IITP-2024-2020-0-01819, ICT Creative Consilience Program, No.2021-0-02068, Artificial Intelligence Innovation Hub).

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

# Appendix

In the following, we describe more comprehensive implementation details, additional analyses, additional experimental results, limitations, future works, and broader impacts of our work.

## A   MORE IMPLEMENTATION DETAILS

Our baseline code is built upon the DenseMatching repository[1]. We implemented the network in PyTorch (Paszke et al., 2019) and used the AdamW optimizer (Loshchilov & Hutter, 2017). All our experiments were conducted on 6 24GB RTX 3090 GPUs. For diffusion reverse sampling, we employed the DDIM sampler (Song et al., 2020a) and set the diffusion timestep $T$ to 5 during both the training and sampling phases. We set the default number of samples for multiple hypotheses to 4 for evaluations on ETH3D and to 3 for HPatches, respectively.

In our experiments, we trained two primary models: the conditional denoising diffusion model and the flow upsampling diffusion model. For the denoising diffusion model, we train 121M modified U-Net based on (Nichol & Dhariwal, 2021) with the learning rate to $1 \times 10^{-4}$ and trained the model for 130,000 iterations with a batch size of 24. For the flow upsampling diffusion model, we used a learning rate of $3 \times 10^{-5}$ and finetuned the pretrained conditional denoising diffusion model for 20,000 iterations with a batch size of 2.

For the feature extraction backbone, we employed VGG-16, as described in Truong et al. (2020b;a). We resized the input images to $H \times W = 512 \times 512$ and extracted feature descriptors at Conv3-3, Conv4-3, Conv5-3, and Conv6-1 with resolutions $\frac{H}{4} \times \frac{W}{4}$, $\frac{H}{8} \times \frac{W}{8}$, $\frac{H}{16} \times \frac{W}{16}$, and $\frac{H}{32} \times \frac{W}{32}$, respectively. We used these feature descriptors to establish both global and local matching costs. The conditional denoising diffusion model was trained at a resolution of 64, while the flow upsampling diffusion model was trained at a resolution of 256 to upsample the flow field from 64 to 256.

## B   EVALUATION DATASETS

We evaluated DiffMatch on standard geometric matching benchmarks: HPatches (Balntas et al., 2017), ETH3D (Schops et al., 2017). To further investigate the effectiveness of the diffusion generative prior, we also evaluated DiffMatch under the harshly corrupted settings (Hendrycks & Dietterich, 2019) of HPatches (Balntas et al., 2017) and ETH3D (Schops et al., 2017). Here, we provide detailed information about these datasets.

**HPatches.** We evaluated our method on the challenging HPatches dataset (Balntas et al., 2017), consisting of 59 image sequences with geometric transformations and significant viewpoint changes. The dataset contains images with resolutions ranging from $450 \times 600$ to $1,613 \times 1,210$.

**ETH3D.** We evaluated our framework on the ETH3D dataset (Schops et al., 2017), which consists of multi-view indoor and outdoor scenes with transformations not constrained to simple homographies. ETH3D comprises images with resolutions ranging from $480 \times 752$ to $514 \times 955$ and consists of 10 image sequences. For a fair comparison, we followed the protocol of (Truong et al., 2020b), which collects pairs of images at different intervals. We selected approximately 500 image pairs from these intervals.

**Corruptions.** Our primary objective is to design a powerful generative prior that can effectively address the inherent ambiguities in dense correspondence tasks, including textureless regions, repetitive patterns, large displacements, or noises. To this end, to assess the robustness of our generative prior against more challenging scenarios, we subjected it to a series of common corruptions from ImageNet-C (Hendrycks & Dietterich, 2019). This benchmark consists of 15 types of algorithmically generated corruptions, which are grouped into four distinct categories: noise, blur, weather, and digital. Each corruption type includes five different severity levels, resulting in a total of 75 unique corruptions. For our evaluation, we specifically focused on severity level 5 to highlight the effectiveness of our generative prior. Note that we use all scenes and rate 15 for the Imagenet-C corrupted versions of HPatches and ETH3D, respectively. In the following, we offer a detailed breakdown of each corruption type.

---

[1]DenseMatching repository: `https://github.com/PruneTruong/DenseMatching`.

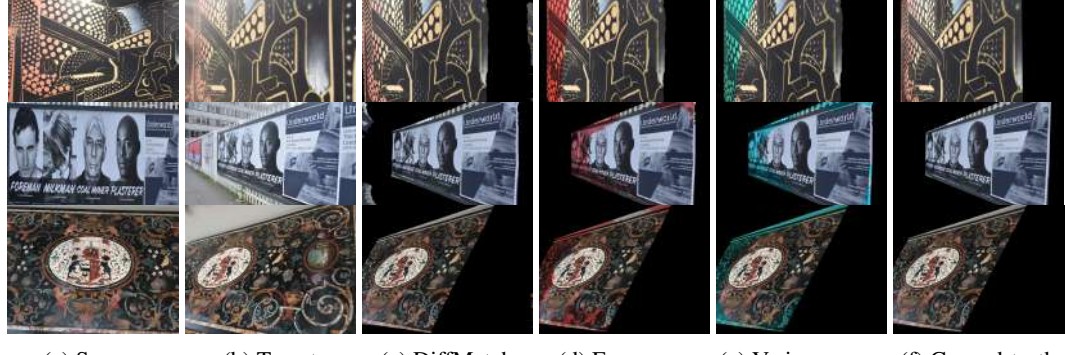

| (a) Source | (b) Target | (c) DiffMatch | (d) Error map | (e) Variance map | (f) Ground-truth |

Figure 6: **Uncertainty estimation.** Our framework can measure the pixel-wise mean and variance of estimated matching fields by sampling from different Gaussian noises. We observe that the variance maps are formed almost the same as the error map, which shows that our variance map successfully expresses the uncertainty of dense correspondence.

Gaussian noise is a specific type of random noise that typically arises in low-light conditions. Shot noise, also known as Poisson noise, is an electronic noise that originates from the inherent discreteness of light. Impulse noise, a color analogue of salt-and-pepper noise, occurs due to bit errors within an image. Defocus blur occurs when an image is out of focus, causing a loss of sharpness. Frosted glass blur is commonly seen on frosted glass surfaces, such as panels or windows. Motion blur arises when the camera moves rapidly, while zoom blur occurs when the camera quickly zooms towards an object. Snow, a type of precipitation, can cause visual obstruction in images. Frost, created when ice crystals form on lenses or windows, can obstruct the view. Fog, which conceals objects and is usually rendered using the diamond-square algorithm, also affects visibility. Brightness is influenced by daylight intensity. Contrast depends on lighting conditions and the color of an object. Elastic transformations apply stretching or contraction to small regions within an image. Pixelation arises when low-resolution images undergo upsampling. JPEG, a lossy image compression format, introduces artifacts during image compression.

## C  ADDITIONAL ANALYSES

### C.1  TRADE-OFF BETWEEN SAMPLING TIME STEPS AND ACCURACY.

Figure 7 illustrates the trade-off between sampling time steps and matching accuracy. As the sampling time steps increase, the matching performance progressively improves in our framework. After time step 5, it outperforms all other existing methods (Truong et al., 2020b;a; Hui et al., 2018; Sun et al., 2018; Hong & Kim, 2021), and the performance also converges. In comparison, DMP (Hong & Kim, 2021), which optimizes the neural network to learn the matching prior of an image pair at test time, requires approximately 300 steps. These results highlight that Diff-Match finds a shorter and better path to accurate matches in relatively fewer steps during the inference phase.

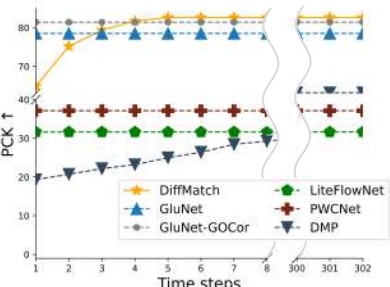

Figure 7: **Time steps vs. PCK.**

### C.2  UNCERTAINTY ESTIMATION

Interestingly, DiffMatch naturally derives the uncertainty of estimated matches by taking advantage of the inherent stochastic property of a generative model. We accomplish this by calculating the pixel-level variance in generated samples across various initializations of Gaussian noise $F_T$. On the other hand, it is crucial to determine when and where to trust estimated matches in dense correspondence (Truong et al., 2021; Kondermann et al., 2008; Mac Aodha et al., 2012; Bruhn & Weickert, 2006; Kybic & Nieuwenhuis, 2011; Wannenwetsch et al., 2017; Ummenhofer et al., 2017). Earlier approaches (Kondermann et al., 2007; 2008; Mac Aodha et al., 2012) relied on post-hoc

Table 6: **Quantitative evaluation on HPatches (Balntas et al., 2017) and ETH3D (Schops et al., 2017) with common corruptions from ImageNet-C (Hendrycks & Dietterich, 2019).** All results are evaluated at corruption severity 5. For simplicity, we denote raw correlation volume and GLU-Net-GOCor as Raw corr. and GOCor (Truong et al., 2020a), respectively. We additionally report the matching performance of the raw correlation volume to demonstrate the effect of our proposed generative matching prior.

| Dataset | Algorithm | Noise | | | Blur | | | | Weather | | | | Digital | | | | Avg. |
|---|---|---|---|---|---|---|---|---|---|---|---|---|---|---|---|---|---|
| | | Gauss. | Shot | Impulse | Defocus | Glass | Motion | Zoom | Snow | Frost | Fog | Bright | Contrast | Elastic | Pixel | JPEG | |
| HPatches | Raw corr. | 156.5 | 149.6 | 153.5 | 104.0 | 94.26 | 244.3 | 176.9 | 227.8 | 254.4 | 222.1 | 104.6 | 141.6 | 116.5 | 197.5 | 131.5 | 165.0 |
| | GLU-Net (Truong et al., 2020b) | 34.96 | 32.60 | 34.18 | 25.74 | 25.71 | 63.26 | 90.75 | 46.16 | 66.63 | 47.81 | 25.28 | 37.45 | 32.85 | 44.31 | 26.94 | 42.31 |
| | GOCor (Truong et al., 2020a) | 27.35 | 27.21 | 26.63 | 23.54 | 20.75 | 57.75 | 88.35 | 39.84 | 63.55 | 36.98 | 21.44 | 23.65 | 28.40 | 33.67 | 22.20 | 36.09 |
| | PDCNet (Truong et al., 2021) | 30.00 | 29.97 | 29.36 | 25.94 | 24.06 | 56.96 | 85.44 | 42.31 | 56.87 | 40.98 | 23.16 | 23.29 | 29.52 | 34.10 | 23.55 | 37.03 |
| | PDCNet+ (Truong et al., 2023) | 29.82 | 27.23 | 28.31 | 21.97 | 19.15 | 48.29 | 81.73 | 35.00 | 82.84 | 35.34 | 17.85 | 21.90 | 27.19 | 33.00 | 22.70 | 35.49 |
| | **DiffMatch** | 31.10 | 28.21 | 29.14 | 21.96 | 19.56 | 38.16 | 97.22 | 37.49 | 50.74 | 35.66 | 20.21 | 27.22 | 27.43 | 37.17 | 21.63 | **34.86** |
| ETH3D | Raw corr. | 103.3 | 94.97 | 102.3 | 41.78 | 36.31 | 141.3 | 135.4 | 153.9 | 177.7 | 140.0 | 50.06 | 60.16 | 62.61 | 95.78 | 63.97 | 97.30 |
| | GLU-Net (Truong et al., 2020b) | 29.20 | 27.51 | 29.11 | 14.18 | 13.16 | 36.90 | 77.73 | 47.11 | 65.22 | 37.75 | 13.89 | 24.52 | 19.11 | 21.25 | 17.77 | 31.63 |
| | GOCor (Truong et al., 2020a) | 27.45 | 25.56 | 26.44 | 11.39 | 10.98 | 33.73 | 73.56 | 43.24 | 66.49 | 39.11 | 10.98 | 18.34 | 15.54 | 16.47 | 15.20 | 28.96 |
| | PDCNet (Truong et al., 2021) | 28.60 | 24.94 | 27.90 | 10.63 | 10.00 | 37.93 | 76.18 | 45.08 | 69.50 | 35.90 | 10.16 | 17.46 | 15.86 | 16.69 | 15.62 | 29.50 |
| | PDCNet+ (Truong et al., 2023) | 30.49 | 26.18 | 28.08 | 8.99 | 8.32 | 33.40 | 68.79 | 39.14 | 65.35 | 31.50 | 8.59 | 8.59 | 14.55 | 14.55 | 13.28 | 27.11 |
| | **DiffMatch** | 25.11 | 23.36 | 24.61 | 8.62 | 5.48 | 36.47 | 72.67 | 41.48 | 64.82 | 25.68 | 8.13 | 15.32 | 12.86 | 17.32 | 14.86 | **26.45** |

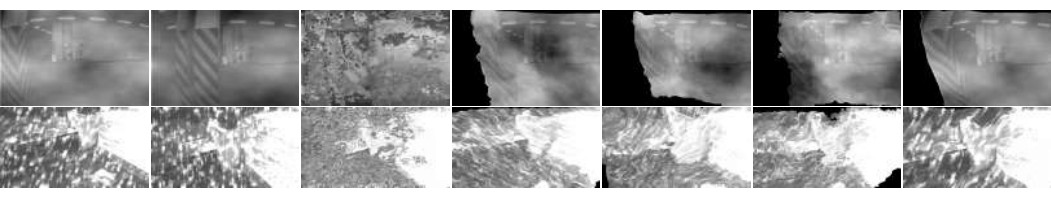

| (a) Source | (b) Target | (c) Raw corr. | (d) GLU-Net | (e) GOCor | (f) PDCNet+ | (g) DiffMatch |
|---|---|---|---|---|---|---|

Figure 8: **Visualizing the effectiveness of the proposed generative matching prior.** The input images are corrupted by fog and snow corruptions (top and bottom, respectively). Compared to raw correlation and previous methods (Truong et al., 2020b;a) that focus solely on point-to-point feature relationships, our approach yields more natural and precise matching results by effectively learning the matching field manifold.

techniques to assess the reliability of models, while more recent model-inherent approaches (Truong et al., 2021; Bruhn & Weickert, 2006; Kybic & Nieuwenhuis, 2011; Wannenwetsch et al., 2017; Ummenhofer et al., 2017) have developed frameworks specifically designed for uncertainty estimation. The trustworthiness of this uncertainty is showcased in Figure 6. We found a direct correspondence between highly erroneous locations and high-variance locations, emphasizing the potential to interpret the variance as uncertainty. We believe this provides promising opportunities for applications demanding high reliability, such as medical imaging (Abi-Nahed et al., 2006) and autonomous driving (Nistér et al., 2004; Chen et al., 2016).

## C.3 THE EFFECTIVENESS OF GENERATIVE MATCHING PRIOR

DiffMatch effectively learns the matching manifold and finds natural and precise matches. In contrast, the raw correlation volume, which is computed by dense scalar products between the source and target descriptors, fails to find accurate point-to-point feature relationships in inherent ambiguities in dense correspondence, including repetitive patterns, textureless regions, large displacements, or noises. To highlight the effectiveness of our generative matching prior, we compare the matching performance evaluated by raw correlation and our method under harshly corrupted settings in Table 6 and Figure 8.

The corruptions introduced by (Hendrycks & Dietterich, 2019) contain the inherent ambiguities in dense correspondence. For instance, snow and frost corruptions obstruct the image pairs by creating repetitive patterns, while fog and brightness corruptions form homogeneous regions. Under these conditions, raw correlation volume fails to find precise point-to-point feature relationships. Conversely, our method effectively finds natural and exact matches within the learned matching manifold, even under severely corrupted conditions. These results highlight the efficacy of our generative prior, which learns both the likelihood and the matching prior, thereby finding the natural matching field even under extreme corruption.

As earlier methods (Pérez et al., 2013; Drulea & Nedevschi, 2011; Werlberger et al., 2010; Lhuillier & Quan, 2000; Liu et al., 2010; Ham et al., 2016) design a hand-crafted prior term as a smoothness

constraint, we can assume that the *smoothness* of the flow field is included in this *prior* knowledge of the matching field. Based on this understanding, we reinterpret Table 6 and Figure 8, showing the comparison between the results of raw correlation and learning-based methods (Truong et al., 2020b;a; 2023; 2021). Previous learning-based approaches predict the matching field with raw correlation between an image pair as a condition. We observe that despite the absence of an explicit prior term in these methods, the qualitative results from them exhibit notably smoother results compared to raw correlation. This difference serves as indicative evidence that the neural network architecture may implicitly learn the matching prior with a large-scale dataset.

However, it is important to note that the concept of *prior* extends beyond mere *smoothness*. This broader understanding underlines the importance of explicitly learning both the data and prior terms simultaneously, as demonstrated in our performance.

### C.4 COMPARISON WITH DIFFUSION-BASED DENSE PREDICTION MODELS

Previous works (Ji et al., 2023; Gu et al., 2022; Saxena et al., 2023b; Duan et al., 2023; Saxena et al., 2023a), applying a diffusion model for dense prediction, such as semantic segmentation (Ji et al., 2023; Gu et al., 2022), or monocular depth estimation (Ji et al., 2023; Duan et al., 2023; Saxena et al., 2023a), use a single RGB image or its feature descriptor as a condition to predict specific dense predictions, such as segmentation or depth map, aligned with the input RGB image. A concurrent study (Saxena et al., 2023a) has applied a diffusion model to predict optical flow, concatenating feature descriptors from both source and target images as input conditions. However, it is

Table 7: **Results via different conditioning schemes.**

| Learning schemes | ETH3D AEPE↓ |
|---|---|
| Feature concat. | 106.83 |
| DiffMatch | **3.12** |

notable that this model is limited to scenarios involving small displacements, typical in optical flow tasks, which differ from the main focus of our study. In contrast, our objective is to predict dense correspondence between two RGB images, source $I_{\text{src}}$ and target $I_{\text{tgt}}$, in more challenging scenarios such as image pairs containing textureless regions, repetitive patterns, large displacements, or noise. To achieve this, we introduce a novel conditioning method which leverages a local cost volume $C^l$ and initial correspondence $F_{\text{init}}$ between two images as conditions, containing the pixel-wise interaction between the given images and the initial guess of dense correspondence, respectively.

To validate the effectiveness of our architecture design, we further train our model using only feature descriptors from source and target, $D_{\text{src}}$ and $D_{\text{tgt}}$, as conditions. This could be a similar architecture design to DDP (Ji et al., 2023) and DDVM (Saxena et al., 2023a), which only condition the feature descriptors from input RGB images. In Table 7, we present quantitative results to compare different conditioning methods and observe that the results with our conditioning method significantly outperform those using two feature descriptors. We believe that the observed results are attributed to the considerable architectural design choice, specifically tailored for dense correspondence.

## D ADDITIONAL RESULTS

### D.1 MORE QUALITATIVE COMPARISON ON HPATCHES AND ETH3D

We provide a more detailed comparison between our method and other state-of-the-art methods on HPatches (Balntas et al., 2017) in Figure 9 and ETH3D (Schops et al., 2017) in Figure 10.

### D.2 MORE QUALITATIVE COMPARISON IN CORRUPTED SETTINGS

We also present a qualitative comparison on corrupted HPatches (Balntas et al., 2017) and ETH3D (Schops et al., 2017) in Figure 11 and Figure 12, respectively.

### D.3 MEGADEPTH

To further evaluate the generalizability of our method, we expanded our evaluation to include the MegaDepth dataset (Li & Snavely, 2018), known for its extensive collection of image pairs exhibiting extreme variations in viewpoint and appearance. Following the procedures used in PDC-Net+ (Truong et al., 2023), we tested our method on 1,600 images.

The quantitative results, presented in Table 8, demonstrate that our approach surpasses PDC-Net+ in performance on the MegaDepth dataset, thereby highlighting the potential for generalizability of our method.

Table 8: **Results on MegaDepth**

| Methods | MegaDepth AEPE ↓ |
|---|---|
| PDC-Net+ (Truong et al., 2021) | 63.97 |
| DiffMatch | **59.73** |

## E    LIMITATIONS AND FUTURE WORK

To the best of our knowledge, we are the first to formulate the dense correspondence task using a generative approach. Through various experiments, we have demonstrated the significance of learning the manifold of matching fields in dense correspondence. However, our method exhibits slightly lower performance on ETH3D (Schops et al., 2017) during intervals with small displacements. We believe this is attributed to the input resolution of our method. Although we introduced the flow upsampling diffusion model, our resolution still remains lower compared to prior works (Truong et al., 2020b;a; 2021; 2023). We conjecture that this limitation could be addressed by adopting a higher resolution and by utilizing inference techniques specifically aimed at detailed dense correspondence, such as zoom-in (Jiang et al., 2021) and patch-match techniques (Barnes et al., 2009; Lee et al., 2021). In future work, we aim to enhance the matching performance by leveraging feature extractors more advanced than VGG-16 (Simonyan & Zisserman, 2014). Moreover, we plan to improve our architectural designs, increase resolution, and incorporate advanced inference techniques to more accurately capture matches.

## F    BROADER IMPACT

Dense correspondence applications have diverse uses, including simultaneous localization and mapping (SLAM)(Durrant-Whyte & Bailey, 2006; Bailey & Durrant-Whyte, 2006), structure from motion (SfM)(Schonberger & Frahm, 2016), image editing (Barnes et al., 2009; Cheng et al., 2010; Zhang et al., 2020), and video analysis (Hu et al., 2018; Lai & Xie, 2019). Although there is no inherent misuse of dense correspondence, it can be misused in image editing to produce doctored images of real people. Such misuse of our techniques can lead to societal problems. We strongly discourage the use of our work for disseminating false information or tarnishing reputations.

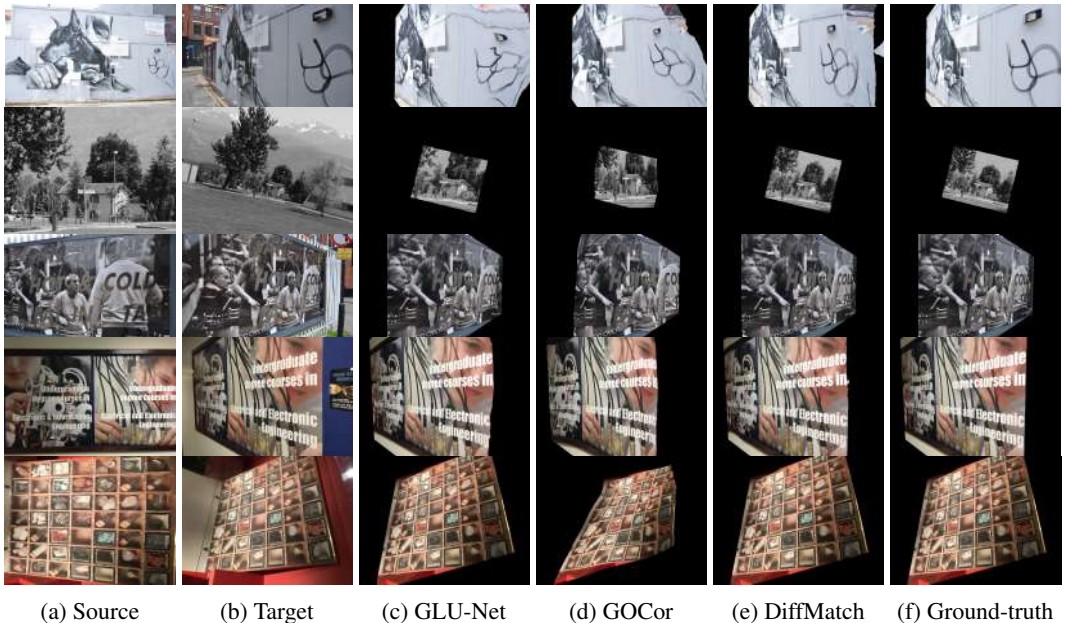

(a) Source     (b) Target     (c) GLU-Net     (d) GOCor     (e) DiffMatch     (f) Ground-truth

Figure 9: **Qualitative results on HPatches (Schops et al., 2017).** The source images are warped to the target images using predicted correspondences.

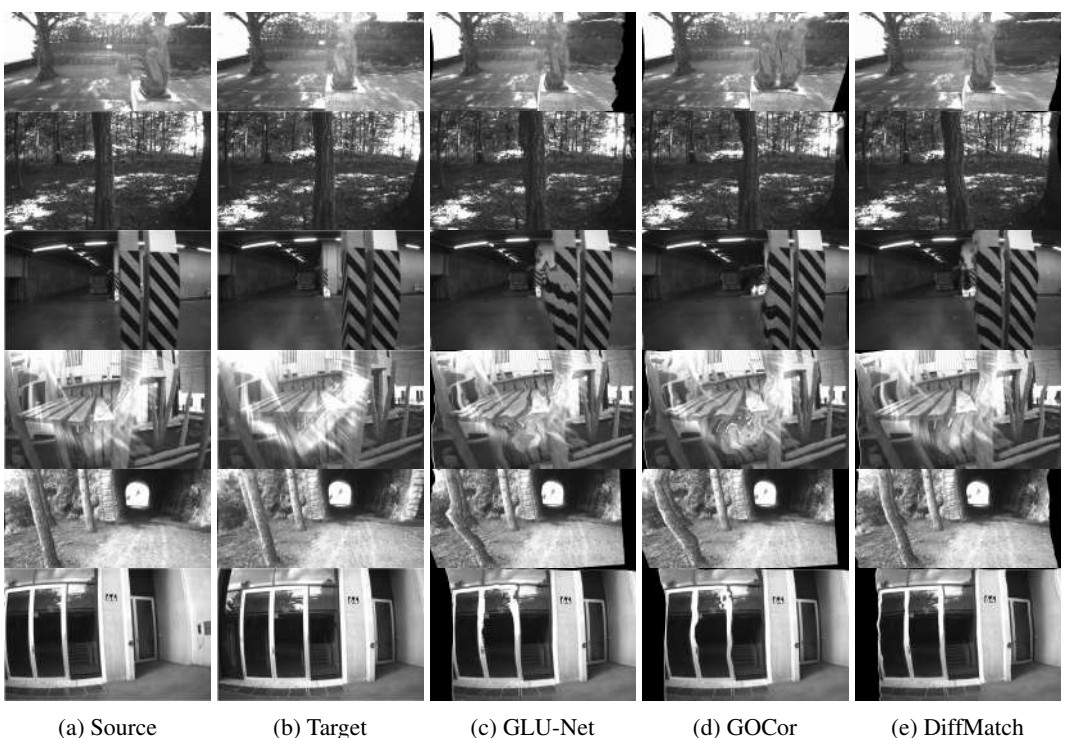

(a) Source     (b) Target     (c) GLU-Net     (d) GOCor     (e) DiffMatch

Figure 10: **Qualitative results on ETH3D (Schops et al., 2017).** The source images are warped to the target images using predicted correspondences.

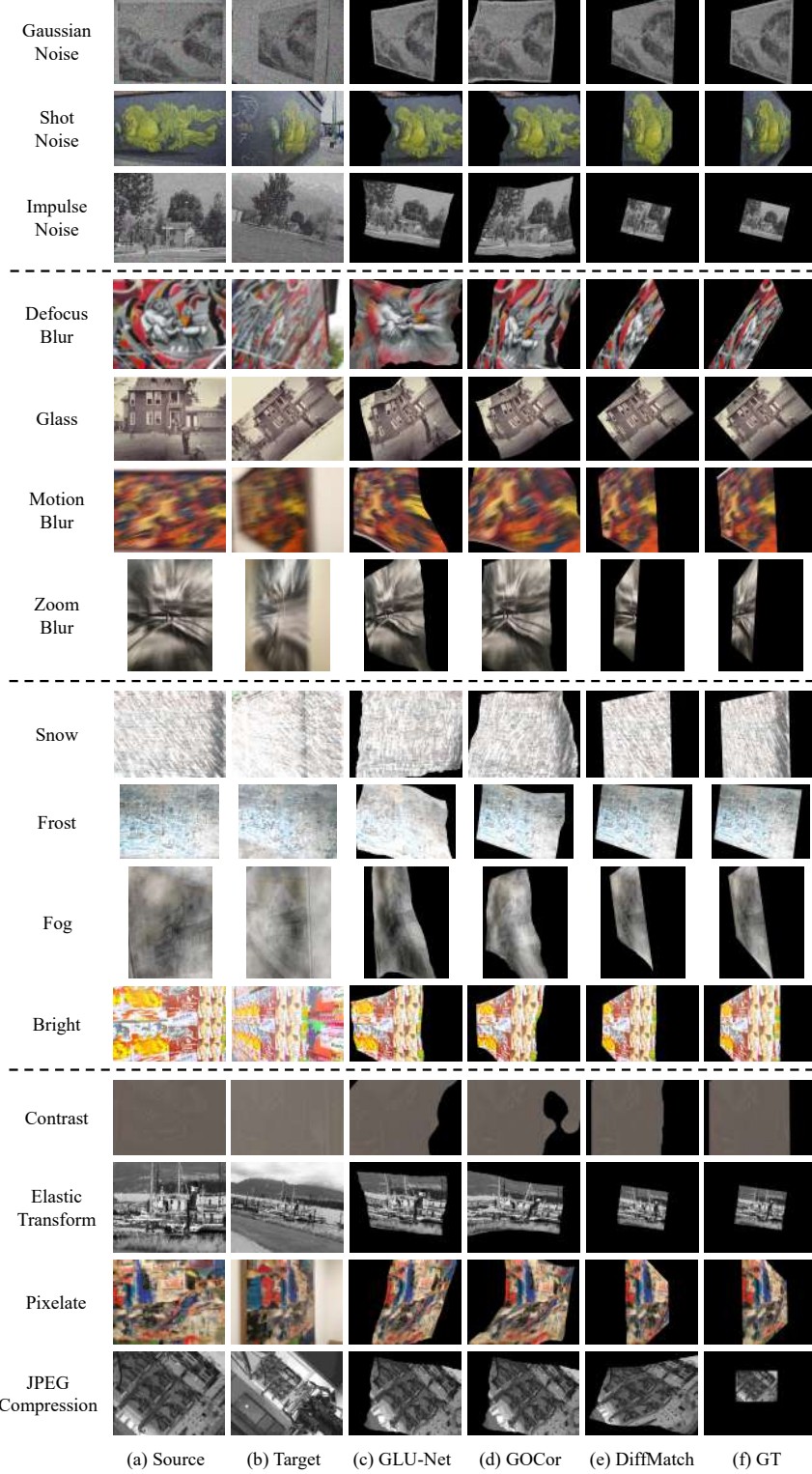

Figure 11: **Qualitative results on HPatches (Schops et al., 2017) using corruptions in Hendrycks & Dieterich (2019).** The source images are warped to the target images using predicted correspondences.

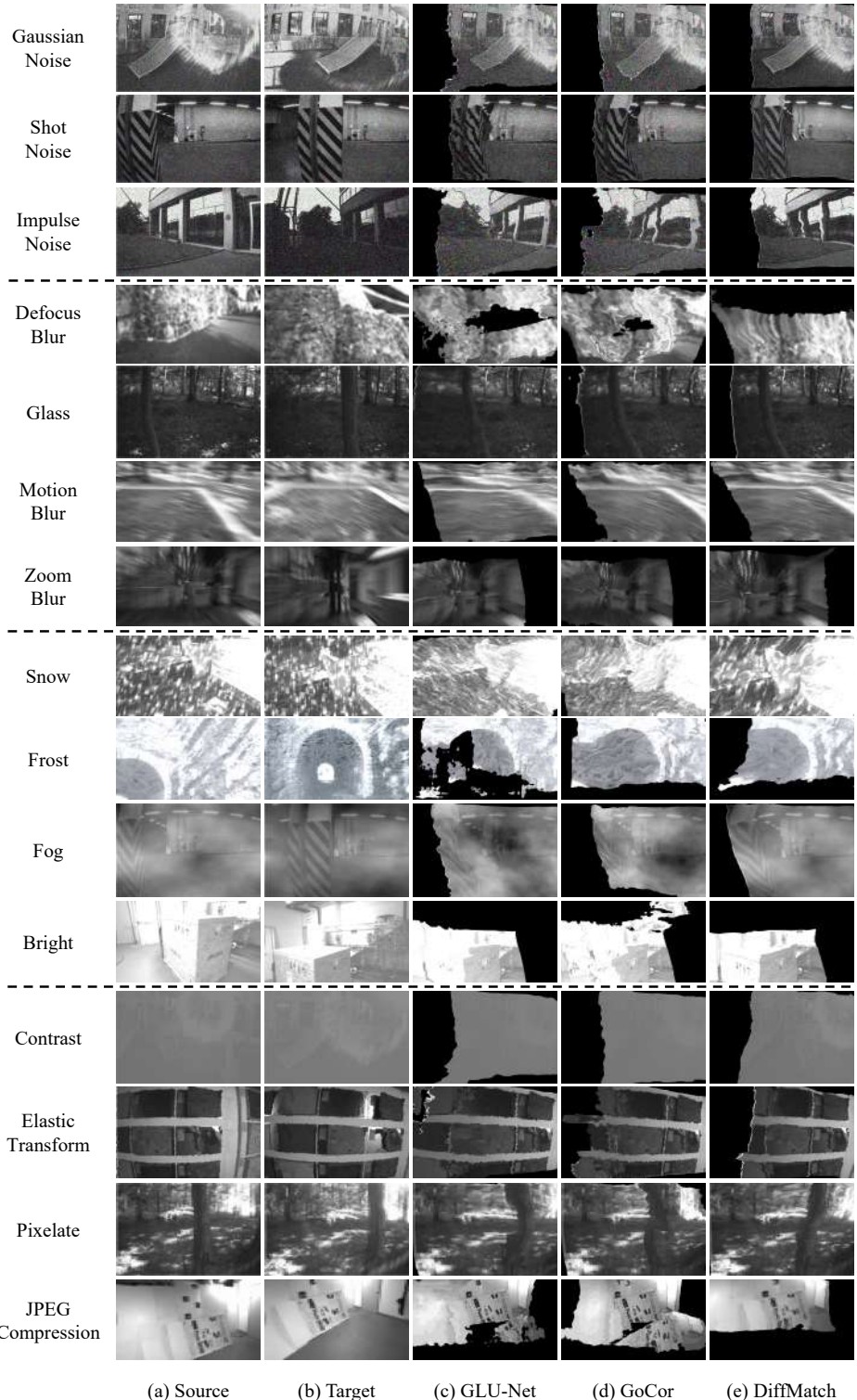

Figure 12: **Qualitative results on ETH3D (Schops et al., 2017) using corruptions in Hendrycks & Dietterich (2019).** The source images are warped to the target images using predicted correspondences.

