# OpenReview forum: "Diffusion Model for Dense Matching"
_ICLR.cc/2024/Conference — ICLR 2024 oral_

### Official Review · Reviewer_tm6R · 2023-10-30

**Soundness:** 4 excellent
**Presentation:** 3 good
**Contribution:** 3 good
**Rating:** 8
**Confidence:** 4

**Summary:**

The authors propose a framework for dense matching for a pair of images using conditional diffusion models. In particular, the framework consists of 2 stages. In the first stage an initial cost volume is computed from the features of the source and target images. The cost volume is used to calculate initial global and local flow information. In the second stage, the initial flow field is refined using a multiscale conditional diffusion model to predict dense correspondence between source and target models. Comparisons are provided with recent baselines and state of the art performance is demonstrated.

**Strengths:**

1. **Paper quality**: The paper is well-written and clearly presented, with attention to detail. The authors have clearly put a lot of effort into making the paper easy to read and understand.
2. **Comparisons**: The paper provides adequate comparisons to several baselines, on two different datasets, demonstrating the effectiveness of the proposed approach.
3. **Ablation**: The ablation studies clearly highlight the need for all of the introduced components, which provides additional evidence for the effectiveness of the proposed approach. The appendix also provides an insightful ablation on the effect of matching quality for number of sampling steps.
4. **Related Works**: An adequate and detailed treatment of related works has been provided to place this work in the context of literature related to dense correspondence computation.
4. **Approach** The proposed approach is simple and elegant, which makes it easy to understand and implement. Represents a good demonstration of the correspondence matching algorithm.
5. **Design of Prior** Computing initial value from a cost volume constructed from a pre-trained VGG-16 network and only learning residual is an efficient strategy as the displacement to be learned is smaller than what would need to be learned otherwise.
6. **Reproducibility** : All the details of the feature extraction network to compute the cost volume and the details of the diffusion model is provided in detail, aiding in the reproducibility of the proposed approach.
6. **Appendix** The authors provide a clear and detailed appendix section, which is helpful for readers who want to learn more about the proposed approach.

**Weaknesses:**

1. **Novelty** : The novelty is somewhat limited to the specific design of the conditioning to the diffusion model. Computation of cost volume using pretrained network have been used in many flow computation networks (as in Glu-Net). In particular, the main novelty is the smart choice of inputs and outputs for the diffusion model. Elaborating a bit more on the challenges of the design will help highlight the novelty of this approach.
3. **Generalization**: The framework shows impressive performance on dense matching for the given datasets, but providing a sense of how generalizable this is to in-the-wild captures, is potentially helpful.
4. **Performance limits**: The qualitative example demonstrated show dense correspondence matching for relative simple transformations between source and target. Providing some insights about how the framework performs for wide baselines or for settings with large viewpoint changes would be helpful.

**Questions:**

1. In this setting is $F_{init}$ extracted from VGG-16 considered the prior term ?
2. What is effect of choice of feature extractor on $F_{init}$. Do features extracted from different extractors like ResNet variants still provide reasonable $F_{init}$ for further optimization? In particular, since this is only used as initialization, would the downstream performance be fairly agnostic to this choice?

---

> ### Author Response · Authors · 2023-11-17
> **Response to reviewer tm6R (Part 1/3)**
>
> ### **General reply**
>
> Thank you for your constructive review and valuable suggestions. Below, we offer detailed responses to each of your questions and comments. If there are any points where our answers don't fully address your concerns, please let us know, and we will respond as quickly as possible.
>
> ---
>
> ### **Weakness 1. Regarding novelty.**
>
> Thank you for your valuable feedback. Firstly, we would like to emphasize that a key novelty of our work is the novel formulation of dense correspondence as a diffusion generative process, a point also positively acknowledged by reviewers **8Vji**, **kgjX**, and **nL9F**. Our approach differs from previous methods [1,2,3] that relied on hand-crafted design prior terms or recent studies [4,5,6] that focus on the data term, assuming that the network architecture learns optimal matching priors with large-scale training datasets. Our unique contribution lies in **learning both the “data” and “prior” terms by leveraging a powerful generative model, the diffusion model**, demonstrating its effectiveness in the challenging dense correspondence task.
>
> Regarding the novelty of our architecture design, we have conducted additional ablation studies compared with previous works [7,8,9,10] that applied diffusion models for specific dense predictions, such as semantic segmentation [7,8] and monocular depth estimation [7,9,10]. These studies typically use a single RGB image or feature descriptor as a condition for predictions aligned with the input RGB image. A concurrent study [11] has also applied a diffusion model to predict optical flow, using concatenated feature descriptors from both source and target images. However, this model is limited to scenarios with small displacements, typically in the optical flow task, which differs from the main focus of our study. DiffMatch, in contrast, aims to predict dense correspondence between two RGB images, $I_\mathrm{src}$ and $I_\mathrm{tgt}$, in scenarios involving textureless regions, repetitive patterns, large displacements, or noise. To achieve this, **we introduced a novel conditioning method using a local cost volume $C^{l}$ and initial flow field $F_\mathrm{init}$**, which captures pixel-wise interactions and initial guesses of dense correspondence between the images.
>
> To validate the effectiveness of our architecture, in response to your insightful suggestion, we trained our model using only feature descriptors from the source and target, $D_\mathrm{src}$ and $D_\mathrm{tgt}$, as conditions. Please note that this approach is similar in architectural design to [7] and [11]. The quantitative results, which compare different conditioning methods, show that our conditioning method significantly outperforms those using only two feature descriptors. We attribute these results to our architectural design, which is specifically tailored for dense correspondence. Table 4 in the “Section 5.3 ABLATION STUDY” of our main paper further demonstrates the effectiveness of each component in our architecture. We will ensure to include an extended discussion and additional ablation studies on this aspect in the final version of our paper. Thank you.
>
> | Conditioning Scheme | ETH3D AEPE↓ |
> | --- | --- |
> | Feature Concat. | 106.83 |
> | DiffMatch | **3.12** |
>
> **Citations:**
>
> [1] Javier Sánchez Pérez, Enric Meinhardt-Llopis, and Gabriele Facciolo. Tv-l1 optical ﬂow estimation. Image Processing On Line, 2013:137–150, 2013.
>
> [2]  Marius Drulea and Sergiu Nedevschi. Total variation regularization of local-global optical ﬂow. ITSC 2011.
>
> [3] Manuel Werlberger, Thomas Pock, and Horst Bischof. Motion estimation with non-local total variation regularization. CVPR 2010.
>
> [4] Seungryong Kim, et al. Fcss: Fully convolutional self-similarity for dense semantic correspondence. CVPR 2017.
>
> [5] Deqing Sun, et al. Pwc-net: Cnns for optical ﬂow using pyramid, warping, and cost volume. CVPR 2018.
>
> [6] Ignacio Rocco, Relja Arandjelovic, and Josef Sivic. Convolutional neural network architecture for geometric matching. CVPR 2017.
>
> [7] Yuanfeng Ji, et al. Ddp: Diffusion model for dense visual prediction. arXiv preprint arXiv:2303.17559, 2023.
>
> [8] Zhangxuan Gu, et al. Diffusioninst: Diffusion model for instance segmentation. arXiv preprint arXiv:2212.02773, 2022.
>
> [9] Saurabh Saxena, et al. Monocular depth estimation using diffusion models, arXiv preprint arXiv:2302.14816, 2023.
>
> [10] Yiqun Duan, Xianda Guo, and Zheng Zhu. Diffusiondepth: Diffusion denoising approach for monocular depth estimation. arXiv preprint arXiv:2303.05021, 2023.
>
> [11] Saurabh Saxena, et al. The surprising effectiveness of diffusion models for optical flow and monocular depth estimation. arXiv preprint arXiv:2306.01923, 2023a.

---

> ### Author Response · Authors · 2023-11-17
> **Response to reviewer tm6R (Part 2/3)**
>
> ### **Weakness 2 & 3. Generalization on other datasets.**
>
> Thank you for your constructive suggestion. First, we would like to clarify that our method is trained by DPED-COCO [1], which consists of augmented synthetic image pairs DPED with randomly and independently moving objects in MS-COCO. This allows our model to alleviate overfitting to synthetic homography and enables our model to be generalized in non-rigid transform.
>
> As you kindly suggest, to address your concerns regarding generalizability, we have extended our evaluation to include the MegaDepth dataset, known for its large-scale collection of image pairs with extreme viewpoint and appearance variations. Following the procedure of PDC-Net+ [2], we tested on 1600 images. The quantitative comparisons below show that our approach outperforms PDC-Net+ on MegaDepth, highlighting the potential of our method for generalizability.
>
> | Methods | MegaDepth AEPE↓ |
> | --- | --- |
> | PDCNet+ | 63.97 |
> | DiffMatch | **59.72** |
>
> Furthermore, we have recently focused on integrating DiffMatch with other dense matching methods to improve its generalizability. This integration is achieved by replacing $F_\mathrm{init}$, derived from the feature backbone (VGG-16 in our case), with the flow field obtained from other models. **We evaluated our method on the semantic matching datasets PF-PASCAL [3] and PF-WILLOW [4]**. We used the state-of-the-art semantic matching model, SD-DINO [5], to determine $F_\mathrm{init}$. Please note that, for a comprehensive experiment, our model ideally should have been trained with SD-DINO. However, due to time constraints during the rebuttal period, we only used the flow field from SD-DINO as $F_\mathrm{init}$ for the pre-trained Conditional Denoising Diffusion Module in the sampling phase. **Interestingly, we found that DiffMatch significantly improves semantic matching performance compared to the original SD-DINO on both PF-PASCAL and PF-WILLOW.** **This indicates the potential of our Conditional Denoising Diffusion Module to be integrated on top of other dense correspondence methods to enhance their performance.** We plan evaluate DiffMatch on additional datasets and explore integrating our model as a plug-in for other models to make it more lightweight. We intend to include these findings in our paper as soon as the experiments are complete. Thank you again for your suggestion.
>
> | Datasets |  | PF-PASCAL |  |  | PF-WILLOW |  |
> | --- | --- | --- | --- | --- | --- | --- |
> | Methods | PCK@0.05 | PCK@0.1 | PCK@0.15 | PCK@0.05 | PCK@0.1 | PCK@0.15 |
> | PDC-Net+ | 34.34 | 56.84 | 70.13 | 30.81 | 54.71 | 68.56 |
> | SD-DINO | **71.67** | 86.04 | 91.92 | 67.26 | 88.61 | 94.32 |
> | DiffMatch combined with SD-DINO | 70.67 | **88.69** | **95.15** | **67.37** | **89.39** | **95.29** |
>
>
> **Citations**:
>
> [1] Prune Truong, et al. Gocor: Bringing globally optimized correspondence volumes into your neural network. NeurIPS 2020.
>
> [2] Prune Truong, et al. Pdc-net+: Enhanced probabilistic dense correspondence network. IEEE PAMI 2023.
>
> [3] Ham, Bumsub, et al. Proposal flow: Semantic correspondences from object proposals. IEEE PAMI 2017.
>
> [4] Ham, Bumsub, et al. Proposal flow. CVPR 2016.
>
> [5] Zhang, Junyi, et al. A Tale of Two Features: Stable Diffusion Complements DINO for Zero-Shot Semantic Correspondence. arXiv preprint arXiv:2305.15347 (2023).

---

> ### Author Response · Authors · 2023-11-17
> **Response to reviewer tm6R (Part 3/3)**
>
> ### **Question 1. Regarding the prior term.**
>
> Thank you for your question. Firstly, we would like to clarify that $F_\mathrm{init}$ is a condition for noise restoration in the Conditional Denoising Diffusion Module, closely related to the “data term” derived from the data, $I_\mathrm{src}$ and $I_\mathrm{tgt}$.
>
> More specifically, in a probabilistic interpretation, the objective for dense correspondence includes a “data” term, which measures matching evidence between source and target features, and a “prior” term, encoding prior knowledge about correspondence. Specifically, the “prior” term in our paper discusses the formation of the matching field manifold and the distribution of the ground-truth flow field.
>
> Traditional methods [1,2,3,4] typically reduce this matching prior to a hand-crafted “smoothness” constraint, suggesting that the flow at one point should be similar to those of its surrounding points. In contrast, current learning-based methods [5,6,7,8] emphasize the “data” term by training neural networks under the assumption that the network architecture itself can learn the optimal matching prior with a large-scale training dataset. However, our paper is the first to propose learning both the “data” and “prior” term simultaneously for dense correspondence, utilizing a diffusion-based generative model. We welcome any further questions you may have. Thank you.
>
> **Citations:**
>
> [1] Javier Sánchez Pérez, Enric Meinhardt-Llopis, and Gabriele Facciolo. Tv-l1 optical ﬂow estimation. Image Processing On Line, 2013:137–150, 2013.
>
> [2] Marius Drulea and Sergiu Nedevschi. Total variation regularization of local-global optical ﬂow. ITSC 2011.
>
> [3] Manuel Werlberger, Thomas Pock, and Horst Bischof. Motion estimation with non-local total variation regularization. CVPR 2010.
>
> [4] Maxime Lhuillier and Long Quan. Robust dense matching using local and global geometric constraints. ICPR 2000.
>
> [5] Seungryong Kim, et al. Fcss: Fully convolutional self-similarity for dense semantic correspondence. CVPR 2017.
>
> [6] Deqing Sun, et al. Pwc-net: Cnns for optical ﬂow using pyramid, warping, and cost volume. CVPR 2018.
>
> [7] Ignacio Rocco, Relja Arandjelovic, and Josef Sivic. Convolutional neural network architecture for geometric matching. CVPR 2017.
>
> [8] Prune Truong, Martin Danelljan, and Radu Timofte. Glu-net: Global-local universal network for dense ﬂow and correspondences. CVPR 2020.
>
> ---
>
> ### **Question 2. Discussion about backbone choice.**
>
> Thank you for your insightful question. As you rightly pointed out, $F_\mathrm{init}$, derived from feature descriptors, plays a crucial role in providing an initial guess within our framework. In response to your suggestion, we tested DiffMatch using an alternative feature backbone—ResNet [1]. It's important to note that our model was initially pre-trained using VGG-16. Ideally, for this analysis, the model should have been trained with ResNet. However, due to the time constraints of the rebuttal period, we simply replaced the $F_\mathrm{init}$ of our VGG-16 pre-trained DiffMatch with a flow field derived from ResNet101 during the sampling phase. **We observed that the initial flow field derived from ResNet feature descriptors was effectively refined by DiffMatch, despite the model not being originally trained with the ResNet backbone.** Nevertheless, it showed lower performance compared to our original results. We believe this is due to two main reasons: firstly, our model was not trained with ResNet, and secondly, the $F_\mathrm{init}$ from ResNet demonstrated lower accuracy than that from VGG-16 in our evaluation dataset, ETH3D. We plan to train our model with different feature backbones, such as ResNet or DINO [2]. We intend to include these results in our paper as soon as the experiments are complete.
>
> |  | ETH3D AEPE↓ |
> | --- | --- |
> |  $F_\mathrm{init}$ by ResNet101 | 16.78 |
> | DiffMatch combined with ResNet101 | 5.74 |
> | $F_{init}​$ by VGG16 | 14.27 |
> | DiffMatch combined with VGG16 | **3.12** |
>
> Moreover, as we addressed in Weaknesses 2 & 3, our findings demonstrate that DiffMatch significantly enhances semantic matching performance on both PF-PASCAL and PF-WILLOW when integrated with SD-DINO. **This suggests the possibility of using other dense matching models as alternative backbones for DiffMatch, potentially enhancing their performance.**
>
> We plan to evaluate DiffMatch on additional datasets and explore the possibility of integrating our model as a plug-in for other models to make it more lightweight. We intend to include these findings in our paper as soon as the experiments are complete. Thank you again for your suggestion.
>
> **Citations:**
>
> [1] He, Kaiming, et al. Deep residual learning for image recognition. CVPR 2016.
>
> [2] Caron, Mathilde, et al. Emerging properties in self-supervised vision transformers. ICCV 2021.

---

> ### Author Response · Authors · 2023-11-23
>
> Dear Reviewer tm6R,
>
> Thank you for the time and effort you have invested in reviewing our paper. We have carefully considered your feedback, included a discussion in the rebuttal, and revised our paper accordingly. As we approach the conclusion of this process, we welcome any additional feedback or suggestions you may have.
>
> Best regards,
>
> The authors of Paper 2370.

---

> > ### Comment · Reviewer_tm6R · 2023-11-23
> > **Response to questions**
> >
> > The authors do a great job of providing detailed responses to each of the concerns raised. I have updated the score to indicate the same.

---

### Official Review · Reviewer_nL9F · 2023-10-30

**Soundness:** 2 fair
**Presentation:** 3 good
**Contribution:** 3 good
**Rating:** 8
**Confidence:** 3

**Summary:**

This work proposes to use a diffusion model to model a data prior for dense correspondence matching. In particular the authors use a standard feature extraction and dense matching stage to have an initial guess for a dense matching field, then they refine it using a diffusion model trained to predict the residual over the initial guess and finally upsample it using a second diffusion model. With this structure they are able to achieve competitive results compared to the SOTA in dense correspondence matching. The use of a diffusion model to model the data prior of their system implies that the proposed solution is actually a generative system that given an initial dense matching field can sample plausible output ones; as a by-product of this formulation the authors propose to model matching uncertainty as discrepancies in the sampling process with some preliminary interesting results.

**Strengths:**

+ **Novel formulation of the problem**: to the best of my knowledge I have not seen a diffusion model used in this context to refine a matching field.

+ **Possibility to model uncertainty**: the proposed formulation models a distribution of plausible matching fields given an initial guess and therefore models implicitly the uncertainty of the matching process. Fig. 6 in the supplementary shows some preliminary analysis of the modeled uncertainty. I found this emerging property of the formulation extremely interesting although only a preliminary exploration is reported in the paper.

**Weaknesses:**

a) **Possible generalization concerns and limited experimental evaluation**: modeling a prior on what a good matching field looks like using a diffusion model exposes the proposed solution to generalization problems since the prior will only model the type of matching flows seen during training. For example in the extreme case where the method is trained only with match fields coming from homographies it will probably not generalize well to other types of non-rigid transformations between frames. The competitors have this type of limitation in a less pronounced way since they focus on improving feature extraction and matching rather than modeling a global prior on what a “good matching field” should look like. Whether this problem arises in practice is hard to estimate from the current paper since the experimental validation is rather limited compared to the main competitors.T he proposed method is evaluated only on two datasets for dense correspondence matching and on two corrupted versions of the same datasets. Competitors like GOCor, PDCNet and PDCNet+ are evaluated on other datasets (e.g., MegaDepth and RobotCar) and additional correspondence tasks (e.g., Optical Flow on KITTI, Pose Estimation on YFCC100M and ScanNet).

b) **Inference time concerns**: Tab. 5 of the paper compares the inference time of PDCNet(+) vs the proposed method with 5 sampling steps and shows that the two proposals are comparable. However in Sec. 4.6 the authors mention that in practice they sample multiple times and average the diffused fields to get the final performance. Depending on how many samples are drawn it will have an impact on the runtime making it grow significantly. From the current paper it is unclear if this multiple sampling strategy is used in the experimental validation ro only in Appendix C.2 and whether the inference time of Tab.5 are taking this into account or not. If not (as it seems like from the text) the inference cost will be significantly higher than competitors.

c) **More ablation studies and unclear dependency on the initialization**: the core of the work is the use of a diffusion model to refine an initial estimation of a matching field ($F_{init}$). From the paper it is unclear how much the prior is able to recover in case of a bad initialization or not and whatever, if possible, the model will need more diffusion steps to recover from a bad conditioning. I would have liked these aspects to be discussed as part of the ablation study. Another interesting ablation that would have nicely complemented the work would have been using the dense cost volume as conditioning to the diffusion process. If the concern is around hardware limitations a test should still be possible at lower resolutions.

**Questions:**

### Questions

1. Can you comment on weakness (b.) and clarify whether the reported numbers are with/without multiple sampling?

2. Could the proposed conditional denoising diffusion module be plugged on top of other dense correspondence methods to enhance their performance (possibly with retraining)? For example could step (b) of this method be combined with PDCNet+ to further boost the performance?


### Suggestions

* Tab. 6 in the appendix is a repetition of Tab. 1 in the main paper
* I would suggest using a more obvious color map for Fig. 6 in the appendix, right now is a bit hard to parse.
* I would also suggest rename DiffMatch to PDCNet+ for the qualitative comparison in Fig. 4-5 since that’s the anime used in the table?

**Details Of Ethics Concerns:**

No concern

---

> ### Author Response · Authors · 2023-11-17
> **Response to reviewer nL9F (Part 1/4)**
>
> ### **General reply**
>
> Thank you for your constructive review and valuable suggestions. Below, we offer detailed responses to each of your questions and comments. If there are any points where our answers don't fully address your concerns, please let us know, and we will respond as quickly as possible.
>
> ---
>
> ### **Weakness a) Possible generalization concerns and limited experimental evaluation.**
>
> Thank you for your valuable suggestion. First, we would like to clarify that our method is trained using DPED-COCO [1], which augments synthetic image pairs from DPED with randomly sampled independently moving objects from MS-COCO. This allows our model to alleviate overfitting to synthetic homography and enables it to generalize to non-rigid transformations.
>
> As you thankfully suggest, we explore other datasets to evaluate the generalizability of our model. We wish to clarify that our method is specifically tailored for challenging dense correspondence tasks, including textureless regions, repetitive patterns, large displacements, or noise. In contrast, the KITTI and RobotCar datasets, which contain road sequences captured by stereo cameras, typically exhibit relatively small displacements and do not align with our primary objective. Similarly, ScanNet and YFCC100M are used for evaluating sparse matching tasks in outdoor and indoor pose estimation, respectively, while our focus is on dense matching. Recent works [2,3] have been evaluated on these datasets, post-processing the results with RANSAC [4]. This was achieved as the goal of these works is to estimate uncertainty of the predicted matches, so that they selectively choose confident matches to find homography by RANSAC.
>
> To address your concerns regarding generalizability, **we have extended our evaluation to include the MegaDepth dataset**, known for its large-scale collection of image pairs with extreme viewpoint and appearance variations. Following the procedure of PDC-Net+ [2], we tested on 1600 images. The quantitative comparisons below show that **our approach outperforms PDC-Net+ on MegaDepth, highlighting the potential of our method for generalizability.**
>
> | Methods | MegaDepth AEPE↓ |
> | --- | --- |
> | PDC-Net+ | 63.97 |
> | DiffMatch | **59.72** |
>
> We plan to evaluate DiffMatch on additional datasets and explore the further generalizability of our method. We intend to include this finding in our paper as soon as the experiments are complete. Additionally, we have recently focused on integrating DiffMatch with other dense matching methods to improve their performance and our generazability. This will be further discussed in our response titled "Response to reviewer nL9F (Part 3/4)." Thank you again for your suggestion.
>
> **Citations:**
>
> [1] Prune Truong, et al. Gocor: Bringing globally optimized correspondence volumes into your neural network. NeurIPS 2020.
>
> [2] Prune Truong, et al. Pdc-net+: Enhanced probabilistic dense correspondence network. IEEE PAMI 2023.
>
> [3] Prune Truong, et al. Learning accurate dense correspondences and when to trust them. CVPR 2021.
>
> [4] Martin A Fischler and Robert C Bolles. Random sample consensus: a paradigm for model fitting with applications to image analysis and automated cartography. Communications of the ACM, 24 (6):381–395, 1981.
>
> [5] Ham, Bumsub, et al. Proposal flow: Semantic correspondences from object proposals. IEEE PAMI 2017.
>
> [6] Ham, Bumsub, et al. Proposal flow. CVPR 2016.
>
> [7] Zhang, Junyi, et al. A Tale of Two Features: Stable Diffusion Complements DINO for Zero-Shot Semantic Correspondence. arXiv preprint arXiv:2305.15347 (2023).

---

> ### Author Response · Authors · 2023-11-17
> **Response to reviewer nL9F (Part 2/4)**
>
> ### **Weakness b) Inference time concerns.**
>
> ### **Question 1. Clarify whether the reported numbers are with/without multiple sampling.**
>
> Thank you for pointing this out. We apologize for omitting the number of samples for multiple hypotheses in Table 5 in “Section 5.3 ABLATION STUDY”. We would like to inform you that in this analysis, we randomly sample 3 Gaussian noises as $F_T$ from an i.i.d. normal distribution for multiple hypotheses. Following your valuable suggestion, we have conducted a more thorough investigation into the trade-off between the number of hypotheses and time consumption. Please note that our framework includes batching the inputs, which significantly mitigates the impact of multiple hypotheses on inference time. Additionally, after addressing several coding issues, we now clearly present quantitative results on the relationship between the number of hypotheses and time efficiency. We observe that varying the number of hypotheses has a minimal effect on inference time, largely thanks to the efficiency gained from input batching. We will make sure to include this corrected table and discussion in the finalized version of our paper. Thanks again for pointing this out.
>
> | Method | C-ETH3D AEPE↓ | Time [ms] |
> | --- | --- | --- |
> | PDC-Net | 29.50 | **112** |
> | PDC-Net+ | 27.11 | **112** |
> | DiffMatch (1 sample, 5 steps) | 27.52 | **112** |
> | DiffMatch (2 sample, 5 steps) | 27.41 | 123 |
> | DiffMatch (3 sample, 5 steps) | **26.45** | 140 |

---

> ### Author Response · Authors · 2023-11-17
> **Response to reviewer nL9F (Part 3/4)**
>
> ### **Weakness c-1) More ablation studies and unclear dependency on the initialization.**
>
> ### **Questions 2. Discussion on integration with other methods.**
>
> Thank you for your insightful question. In the Appendix, Table 6 and Figure 8 in “Section C.3 THE EFFECTIVENESS OF GENERATIVE MATCHING PRIOR” present a comparison between the initial flow field $F_\mathrm{init}$, derived from raw correlation, and the refined flow field resulting from DiffMatch on the harshly corrupted HPatches and ETH3D. While the raw correlation results in poor performance, with Average Endpoint Errors (AEPEs) of 165.0 and 96.30 respectively, DiffMatch effectively refines this poor initialization using the diffusion generative prior, achieving significantly better performance with AEPEs of 33.15 and 26.45, respectively.
>
> Furthermore, as you insightfully suggested, considering the relationship between the initial flow and overall performance, there seems to be substantial potential for performance enhancement by using results from other models as the initial flow field. We appreciate your excellent suggestion. One issue under consideration is the need to extract the results of existing models for training. Due to time constraints, we were unable to conduct this experiment. However, we chose a simpler approach which replaces $F_\mathrm{init}$ in our VGG-16 pre-trained model with the flow field obtained from a pre-trained PDC-Net+. Unfortunately, as shown in the below table, we did not find any improvement over the performance of our paper. We think this is because the flow field from PDC-Net+ differs significantly from the VGG-16 $F_\mathrm{init}$ used in the training phase. The flow field from PDC-Net+ tends to produce much smoother results compared to the flow field from raw correlation, as PDC-Net+ learns smoothness in their network with a large-scale dataset.
>
> |  | ETH3D AEPE↓ | C-ETH3D AEPE↓ |
> | --- | --- | --- |
> | DiffMatch  | **3.12** | **26.45** |
> | PDC-Net+  | 3.14 | 27.11 |
> | DiffMatch combined with PDC-Net+  | 3.55 | 26.49 |
>
> Instead, we evaluated our method on the semantic matching datasets PF-PASCAL [1] and PF-WILLOW [2]. We used the state-of-the-art semantic matching model, SD-DINO [3], to determine $F_\mathrm{init}$. Please note that SD-DINO uses intermediate diffusion features in a pre-trained diffusion model, so there is no need for training or fine-tuning. We believe that not training the neural network and directly using the feature descriptors from the diffusion model seems to provide more adequate initial flow field for our network. **Interestingly, we found that DiffMatch notably improves semantic matching performance compared to the original SD-DINO on both PF-PASCAL and PF-WILLOW.** **This indicates the potential of our Conditional Denoising Diffusion Module to be integrated on top of other dense correspondence methods to enhance their performance.**
>
> | Datasets |  | PF-PASCAL |  |  | PF-WILLOW |  |
> | --- | --- | --- | --- | --- | --- | --- |
> | Methods | PCK@0.05 | PCK@0.1 | PCK@0.15 | PCK@0.05 | PCK@0.1 | PCK@0.15 |
> | PDC-Net+ | 34.34 | 56.84 | 70.13 | 30.81 | 54.71 | 68.56 |
> | SD-DINO | **71.67** | 86.04 | 91.92 | 67.26 | 88.61 | 94.32 |
> | DiffMatch combined with SD-DINO | 70.67 | **88.69** | **95.15** | **67.37** | **89.39** | **95.29** |
>
> We find your suggestion very promising and plan to explore integrating our model as a plug-in for other models to make it more lightweight. We intend to include these findings in our paper as soon as the experiments are complete. Thank you again for your suggestion.
>
> **Citations:**
>
> [1] Ham, Bumsub, et al. Proposal flow: Semantic correspondences from object proposals. IEEE PAMI 2017.
>
> [2] Ham, Bumsub, et al. Proposal flow. CVPR 2016.
>
> [3] Zhang, Junyi, et al. A Tale of Two Features: Stable Diffusion Complements DINO for Zero-Shot Semantic Correspondence. arXiv preprint arXiv:2305.15347 (2023).
>
> ---
>
> ### **Weakness c-2) Regarding the global cost volume as a condition.**
>
> Thank you for your insightful suggestion. As you mentioned, leveraging a global cost volume might indeed provide the model with pixel-wise interaction between given images, potentially enhancing performance. However, as you rightly pointed out, we have empirically found that the increased memory consumption required to build the cost volume, along with the increased channel dimension, pose significant challenges. To address this, as you kindly proposed, utilizing a global cost volume at a lower resolution could be an effective strategy to establish a coarse flow field. We are planning to conduct experiments that leverage a coarsely estimated flow field, derived from a global cost volume, as the initial flow field instead of those derived from raw cost volume. This approach, we anticipate, could further improve the overall performance of our model by providing a better initialization. We will include these findings in our paper as soon as the experiments are complete.

---

> ### Author Response · Authors · 2023-11-17
> **Response to reviewer nL9F (Part 4/4)**
>
> ### **Suggestions**
>
> Thank you for pointing these out.
>
> - We acknowledge that Table 6 in the appendix partly repeats Table 1 in the main paper. Following your suggestion, we will revise Table 6 to focus on the comparison between results of raw correlation and current learning-based methods.
> - We also agree that a clearer color map for Figure 6 is necessary for better readability and will implement this change.
> - Furthermore, for consistency in terminology, we will include the qualitative results of PDC-Net+ in Figures 4-5. We are grateful for your suggestions, which will undoubtedly enhance the quality and coherence of our paper.
>
> We will make sure all your suggestions are included in the revised version of our paper.

---

> > ### Comment · Reviewer_nL9F · 2023-11-22
> > **Post rebuttal comment**
> >
> > Thank you for the detailed answer, I think the new results address most of my concerns and therefore I will raise my score for the paper.

---

### Official Review · Reviewer_kgjX · 2023-11-03

**Soundness:** 4 excellent
**Presentation:** 3 good
**Contribution:** 4 excellent
**Rating:** 8
**Confidence:** 3

**Summary:**

This paper proposes a new technique for finding dense correspondences between two 2D RGB images, using a generative rather than discriminative model, specifically a conditional diffusion model. The key insight is that this allows optimizing the full posterior (data and prior terms in the Bayesian formulation) instead of the likelihood. The authors propose additional technical components to get the pipeline to work robustly and accurately.

**Strengths:**

With the caveat that this is not my precise area of specialization: I enjoyed reading the paper and think that the proposed method is elegant and interesting. The idea of treating the correspondence field as an image to be synthesized is compelling. The additional components in the pipeline (e.g. for super-res) seem appropriately chosen. The results are good -- even if they don't always beat state-of-the-art baselines -- and definitely good enough given that the technique is of independent methodological interest.

**Weaknesses:**

"These approaches assume that the matching prior can be learned within the model architecture by leveraging the high capacity of deep networks"

For the argument in the paper to be more compelling, the above statement needs to be clarified. Exactly how is the prior "learned within the model architecture"? Can we say something more precise about how the prior is captured, and how much of it, in these earlier methods?

How many samples were used to compute the MAP estimates used for statistics in the tables, as per Section 4.6? And were these samples chosen i.i.d. from the standard normal distribution?

**Questions:**

Minor:
- Please don't write $1e^{-4}$ when (I assume) you mean $10^{-4}$. $e$ is the base of natural logarithms. If you must use scientific number formats (please do it only in code, not papers!), do note that it's written $1e-4$, not $1e^{-4}$.

---

> ### Author Response · Authors · 2023-11-17
> **Response to reviewer kgjX (Part 1/2)**
>
> ### **General reply**
>
> Thank you for your constructive review and valuable suggestions. Below, we offer detailed responses to each of your questions and comments. If there are any points where our answers don't fully address your concerns, please let us know, and we will respond as quickly as possible.
>
> ---
>
> ### **Weakness 1. Discussion about matching prior.**
> > Exactly how is the prior "learned within the model architecture"?
>
> Thank you for your constructive question. We would like to clarify that the concept of “the matching prior within the model architecture” has been already discussed in previous works [1,2,3]. These studies argue that the matching prior inherently exists within the model architecture or is learned from large-scale training datasets. However, as you insightfully asked, “how the prior is captured in the model architecture and how much it is” might be challenging to precisely visualize, because we regard the “prior term” as encoding the general prior knowledge of correspondence and matching field manifold.
>
> As earlier methods [4,5,6] design a hand-crafted prior term as a smoothness constraint, we can assume that the "smoothness" of the flow field is included in this "prior" knowledge of the matching field. Based on this understanding, we reinterpret Table 6 and Figure 8 in "Section C.3 THE EFFECTIVENESS OF GENERATIVE MATCHING PRIOR", showing the comparison between the results of raw correlation and learning-based methods [7,8,9]. Previous learning-based approaches predict the matching field with raw correlation between an image pair as a condition. We observe that despite the absence of an explicit prior term in these methods, the qualitative results from them exhibit notably smoother results compared to raw correlation. This difference serves as indicative evidence that the neural network architecture may implicitly learn the matching prior with a large-scale dataset.
>
> However, it is important to note that the concept of "prior" extends beyond mere "smoothness”. This broader understanding underlines the importance of explicitly learning both the data and prior terms simultaneously, as proposed in our paper. We will make sure to include this extended discussion in the final version of our paper. Thank you for pointing this out.
>
> **Citations:**
>
> [1] Dmitry Ulyanov, Andrea Vedaldi, and Victor Lempitsky. Deep image prior. CVPR 2018.
>
> [2] Alexey Dosovitskiy and Thomas Brox. Inverting visual representations with convolutional networks. CVPR 2016.
>
> [3] Sunghwan Hong and Seungryong Kim. Deep matching prior: Test-time optimization for dense correspondence. ICCV 2021.
>
> [4] Javier Sánchez Pérez, Enric Meinhardt-Llopis, and Gabriele Facciolo. Tv-l1 optical ﬂow estimation. Image Processing On Line, 2013:137–150, 2013.
>
> [5] Marius Drulea and Sergiu Nedevschi. Total variation regularization of local-global optical ﬂow. ITSC 2011.
>
> [6] Manuel Werlberger, Thomas Pock, and Horst Bischof. Motion estimation with non-local total variation regularization. CVPR 2010.
>
> [7] Seungryong Kim, Dongbo Min, Bumsub Ham, Sangryul Jeon, Stephen Lin, and Kwanghoon Sohn. Fcss: Fully convolutional self-similarity for dense semantic correspondence. CVPR 2017.
>
> [8] Deqing Sun, Xiaodong Yang, Ming-Yu Liu, and Jan Kautz. Pwc-net: Cnns for optical ﬂow using pyramid, warping, and cost volume CVPR 2018.
>
> [9] Ignacio Rocco, Relja Arandjelovic, and Josef Sivic. Convolutional neural network architecture for geometric matching. CVPR 2017.

---

> ### Author Response · Authors · 2023-11-17
> **Response to reviewer kgjX (Part 2/2)**
>
> ### **Weakness 2. Regarding multiple hypotheses.**
> > How many samples were used to compute the MAP estimates used for statistics in the tables, as per Section 4.6?
>
> Thank you for pointing this out. We would like to inform you that, by default, we randomly sample 3 Gaussian noises as $F_T$ from an i.i.d. normal distribution for multiple hypotheses. Furthermore, we provide a quantitative comparison between different numbers of samples below. We found that as the number of samples increases, the performance becomes more stable and improves, since more samples contribute to a more accurate distribution of the matching field. We will include the number of samples in the final version of our paper and add this ablation study in the final version of the Appendix. Thank you for pointing this out again.
>
> | Method | C-ETH3D AEPE↓ | Time [ms] |
> | --- | --- | --- |
> | PDC-Net | 29.50 | **112** |
> | PDC-Net+ | 27.11 | **112** |
> | DiffMatch (1 sample, 5 steps) | 27.52 | **112** |
> | DiffMatch (2 sample, 5 steps) | 27.41 | 123 |
> | DiffMatch (3 sample, 5 steps) | **26.45** | 140 |
>
> ---
>
> ### **Question 1. Regarding notation.**
>
> Thank you for pointing this out. We will fix our notations according to your suggestion in the revised version of our paper.

---

> > ### Comment · Reviewer_kgjX · 2023-11-23
> > **Ok with acceptance**
> >
> > Dear authors,
> >
> > Thank you for your comments. I definitely endorse including the expanded discussion re priors in the final revision. I'll continue to support acceptance and maybe raise my score for clarity.

---

### Official Review · Reviewer_8Vji · 2023-11-04

**Soundness:** 4 excellent
**Presentation:** 4 excellent
**Contribution:** 3 good
**Rating:** 8
**Confidence:** 4

**Summary:**

The paper presents DiffMatch, a novel framework for dense matching that explicitly models both the data and prior terms. DiffMatch leverages a conditional denoising diffusion model to address inherent ambiguities of matching, resulting in significant performance improvements over existing techniques. The paper argues that recent approaches have focused on learning the data term with deep neural networks without explicitly modeling the prior, but these approaches often fail to address inherent ambiguities of matching. DiffMatch addresses these issues by explicitly modeling both the data and prior terms using a diffusion model, which is trained to denoise the input image conditioned on the output of the matching network. The paper provides experimental results demonstrating the effectiveness of DiffMatch on several benchmark datasets, achieving state-of-the-art performance in terms of accuracy and efficiency.

**Strengths:**

The paper presents a novel framework for dense matching, DiffMatch, and shows significant performance improvements over existing techniques. I believe this is one of the first work to apply diffusion model to solve dense correspondence (flow estimation) tasks and the results are very encouraging. The proposed approach tries to address inherent ambiguities of matching, such as textureless regions, repetitive patterns, large displacements, or noises. The approach also seems to be efficient and scalable, making it suitable for real-world applications. Overall, the paper's contributions are significant in advancing the field of dense matching.

The paper is well-written and clearly demonstrates the proposed approach and experimental results. The authors provide a detailed explanation of the diffusion model and its application to dense matching, as well as a thorough evaluation of the proposed approach on several benchmark datasets. The paper's contributions are supported by the experimental results, which demonstrate the effectiveness of the proposed approach. The paper is well-organized and easy to follow. The authors provide a clear explanation of the proposed approach and its application to dense matching, as well as a detailed evaluation of the approach on several benchmark datasets. The paper's contributions are clearly presented and supported by the experimental results.

**Weaknesses:**

I do not have major concerns on the paper less lacking some details. One notable improvement will be adding more discussions to diffusion based dense prediction networks, especially methods like DDP [1]. It is questionable to me why DDP is not directly applicable to the task of dense matching. Another possible improvement is to add diffusion-based dense prediction models as baselines to the method (\eg a DDP model trained on dense flow supervision).

**Questions:**

1. It is common for dense matching models to also test on various tasks such as optical flow estimation (KITTI) and two-view geometry estimation (ScanNet / YFCC100M). Are there specific reason your model cannot achieve these similar tasks?
2. Adding more discussions / baselines to some diffusion-enabled dense prediction networks would further strengthen my recommendation.

---

> ### Author Response · Authors · 2023-11-17
> **Response to reviewer 8Vji (Part 1/2)**
>
> ### **General reply**
>
> Thank you for your constructive review and valuable suggestions. Below, we offer detailed responses to each of your questions and comments. If there are any points where our answers don't fully address your concerns, please let us know, and we will respond as quickly as possible.
>
> ---
>
> ### **Weakness. Comparison with diffusion-based dense prediction models.**
>
> > It is questionable to me why DDP is not directly applicable to the task of dense matching.
>
>
> ### **Question 2. More discussions or baselines about some diffusion-enabled dense prediction networks.**
>
> > Adding more discussions / baselines to some diffusion-enabled dense prediction networks.
>
>
> Thank you for your question. First, we wish to clarify that the previous methods [1,2,3,4] applying a diffusion model for dense prediction, such as semantic segmentation [1,2], or monocular depth estimation [1,3,4], use a single RGB image or its feature descriptor as a condition to predict specific dense predictions, such as segmentation or depth map, aligned with the input RGB image. A concurrent study [5] has applied a diffusion model to predict optical flow, concatenating feature descriptors from both source and target images as input conditions. However, it is notable that this model is limited to scenarios involving small displacements, typical in optical flow tasks, which differ from the main focus of our study. In contrast, our objective is to predict dense correspondence between two RGB images, source $I_\mathrm{src}$ and target $I_\mathrm{tgt}$, in more challenging scenarios such as image pairs containing textureless regions, repetitive patterns, large displacements, or noise. To achieve this, we introduce a novel conditioning method which leverages a local cost volume $C^{l}$ and initial flow field $F_\mathrm{init}$ between two images as conditions, containing the pixel-wise interaction between the given images and the initial guess of dense correspondence, respectively.
>
> To validate the effectiveness of our architecture design, as you have thankfully suggested, we train our model using only feature descriptors from source and target, $D_\mathrm{src}$ and $D_\mathrm{tgt}$, as conditions. Please note that this method could be a similar architecture design to DDP [1] and DDVM [5], which only condition the feature descriptors from input RGB images. We present quantitative results to compare different conditioning methods and observe that the results with our conditioning method significantly outperform those using two feature descriptors. We believe that the observed results are attributed to the considerable architectural design choice, specifically tailored for dense correspondence. Table 4 in "Section 5.3 ABLATION STUDY" of the main paper also shows the effectiveness of each component in our architecture. We will make sure to include this extended discussion and ablation studies on this aspect in the final version of our paper. Thank you for pointing this out.
>
> | Conditioning Scheme | ETH3D AEPE↓ |
> | --- | --- |
> | Feature Concat | 106.83 |
> | DiffMatch | **3.12** |
>
> **Citations:**
>
> [1] Yuanfeng Ji, et al. Ddp: Diffusion model for dense visual prediction. arXiv preprint arXiv:2303.17559, 2023.
>
> [2] Zhangxuan Gu, et al. Diffusioninst: Diffusion model for instance segmentation. arXiv preprint arXiv:2212.02773, 2022.
>
> [3] Saurabh Saxena, et al. Moncular depth estimation using diffusion models. arXiv preprint arXiv:2302.14816, 2023.
>
> [4] Yiqun Duan, Xianda Guo, and Zheng Zhu. Diffusiondepth: Diffusion denoising approach for monocular depth estimation. arXiv preprint arXiv:2303.05021, 2023.
>
> [5] Saurabh Saxena, et al. The surprising effectiveness of diffusion models for optical flow and monocular depth estimation. arXiv preprint arXiv:2306.01923, 2023a.

---

> ### Author Response · Authors · 2023-11-17
> **Response to reviewer 8Vji (Part 2/2)**
>
> ### **Question 1. Testing on other datasets.**
>
> > Are there specific reason your model cannot achieve these similar tasks?
>
>
> Thank you for your valuable suggestion. We wish to clarify that our method is specifically tailored for challenging dense correspondence tasks, including textureless regions, repetitive patterns, large displacements, or noise. In contrast, the KITTI dataset, which contains road sequences captured by stereo cameras, typically exhibits relatively small displacements and does not align with our primary objective. Similarly, ScanNet and YFCC100M are used for evaluating sparse matching tasks in outdoor and indoor pose estimation, respectively, while our focus is on dense matching. Recent works [1,2] evaluated on these datasets, by post-processing the results with RANSAC [3]. This was achieved as the goal of these works is to estimate uncertainty of the predicted matches, so that they selectively choose confident matches to find homography by RANSAC.
>
> To address your concerns regarding generalizability, **we have extended our evaluation to include the MegaDepth dataset, known for its large-scale collection of image pairs with extreme viewpoint and appearance variations.** Following the procedure of PDC-Net+ [2], we tested on 1600 images. The quantitative comparisons below show that **our approach outperforms PDC-Net+ on MegaDepth, highlighting the potential of our method for generalizability.**
>
> | Methods | MegaDepth AEPE↓ |
> | --- | --- |
> | PDC-Net+ | 63.97 |
> | DiffMatch | **59.73** |
>
> Furthermore, we have recently focused on integrating DiffMatch with other dense matching methods to improve its generalizability. This integration is achieved by replacing $F_\mathrm{init}$, derived from the feature backbone (VGG-16 in our case), with the flow field obtained from other models. **We evaluated our method on the semantic matching datasets PF-PASCAL [4] and PF-WILLOW [5]**. We used the state-of-the-art semantic matching model, SD-DINO [6], to determine $F_\mathrm{init}$. Please note that, for a comprehensive experiment, our model ideally should have been trained with SD-DINO. However, due to time constraints during the rebuttal period, we only used the flow field from SD-DINO as $F_\mathrm{init}$ for the pre-trained Conditional Denoising Diffusion Module in the sampling phase. **Interestingly, we found that DiffMatch significantly improves semantic matching performance compared to the original SD-DINO on both PF-PASCAL and PF-WILLOW.** **This indicates the potential of our Conditional Denoising Diffusion Module to be integrated on top of other dense correspondence methods to enhance their performance.**
>
> We plan to evaluate DiffMatch on additional datasets and explore the possibility of integrating our model as a plug-in for other models to make it more lightweight. We intend to include these findings in our paper as soon as the experiments are complete. Thank you again for your suggestion.
>
> | Datasets |  | PF-PASCAL |  |  | PF-WILLOW |  |
> | --- | --- | --- | --- | --- | --- | --- |
> | Methods | PCK@0.05 | PCK@0.1 | PCK@0.15 | PCK@0.05 | PCK@0.1 | PCK@0.15 |
> | PDC-Net+ | 34.34 | 56.84 | 70.13 | 30.81 | 54.71 | 68.56 |
> | SD-DINO | **71.67** | 86.04 | 91.92 | 67.26 | 88.61 | 94.32 |
> | DiffMatch combined with SD-DINO | 70.67 | **88.69** | **95.15** | **67.37** | **89.39** | **95.29** |
>
>
> **Citations:**
>
> [1] Prune Truong, et al. Learning accurate dense correspondences and when to trust them. CVPR 2021.
>
> [2] Prune Truong, et al. Pdc-net+: Enhanced probabilistic dense correspondence network. IEEE PAMI, 2023.
>
> [3] Martin A Fischler and Robert C Bolles. Random sample consensus: a paradigm for model fitting with applications to image analysis and automated cartography. Communications of the ACM, 24 (6):381–395, 1981.
>
> [4] Ham, Bumsub, et al. Proposal flow: Semantic correspondences from object proposals. IEEE PAMI 2017.
>
> [5] Ham, Bumsub, et al. Proposal flow. CVPR 2016.
>
> [6] Zhang, Junyi, et al. A Tale of Two Features: Stable Diffusion Complements DINO for Zero-Shot Semantic Correspondence. arXiv preprint arXiv:2305.15347 (2023).

---

> > ### Comment · Reviewer_8Vji · 2023-11-22
> >
> > Thanks for these interesting results. I'm raising my scores since all my concerns have been addressed.

---

### Author Response · Authors · 2023-11-17
**General Response to Reviewers**

We thank the reviewers for their praise regarding the writing of our paper (**8Vji**, **kgjX**, **nL9F**, **tm6R**), and the novel formulation of the problem, which first applies a diffusion model to dense correspondence (**8Vji**, **kgjX**, **nL9F**). We also value their appreciation for our comprehensive experiments and analyses (**8Vji**, **tm6R**) and their recognition of the efficiency and scalability of our approach (**8Vji**, **tm6R**).

---

### Author Response · Authors · 2023-11-21
**General Response**

Dear reviewers,

Thank you once again for dedicating time to review our paper and for valuable contributions towards improving it. As the discussion period is nearing its conclusion, we would like to kindly invite the reviewers to check our responses.

****8Vji:**** We greatly appreciate the reviewer's constructive feedback, which includes the suggestion to discuss diffusion-based dense prediction models and test on additional datasets to better evaluate the generalizability of our method. In response, we have enriched our rebuttal by comparing our method with existing diffusion-based dense prediction models and have conducted further ablation studies to assess the efficacy of our conditioning scheme. Additionally, we have extended our evaluation to include the real-world dataset MegaDepth, as well as the semantic matching datasets PF-PASCAL and PF-WILLOW, thereby addressing concerns regarding generalizability. We sincerely hope that the reviewer will recognize the depth of our investigation and contribute further to this enriching discussion.

****kgjX:**** We thank the reviewer for the thorough review regarding the discussion of the matching prior and multiple hypotheses. In the rebuttal, we have included a detailed discussion on “how the prior is learned within the model architecture” and have reinterpreted “Section C.3 THE EFFECTIVENESS OF GENERATIVE MATCHING PRIOR” in the Appendix, focusing on the aspect of the matching prior. Additionally, we have incorporated detailed information about multiple hypotheses and conducted further ablation studies to explore the relationship between the number of hypotheses, matching accuracy, and time complexity. We earnestly hope that our responses adequately address your questions and invite the reviewer to further contribute to this discussion.

****nL9F:**** We appreciate the reviewer’s detailed review of our paper, particularly regarding testing our method on different datasets, concerns about inference time, more ablations on initialization, discussions on integration with other methods, and leveraging global cost as conditions. Addressing all the reviewers' comments, we have included additional evaluations of our method on the real-world dataset MegaDepth and the semantic matching datasets PF-PASCAL and PF-WILLOW, thus addressing concerns about the generalizability of our method. We have also provided a clear presentation on the relationship between inference time complexity and performance, addressing concerns about inference time. Following the reviewer’s suggestion, we further explored and demonstrated the potential of our method as a plug-in for other models. Moreover, we have included discussions about incorporating global cost volume as conditions in our framework. We sincerely hope that the reviewer will contribute to this enriched discussion and check the comprehensive investigations we have undertaken.

****tm6R:**** We thank the reviewer for the insightful reviews concerning the novelty of our paper, its generalizability on in-the-wild datasets, and the choice of backbone. In response to the comments, we have included additional discussion on the novelty of our paper, which firstly reformulates dense correspondence as a diffusion process and further elaborates on the architectural novelty with additional ablation studies. We have also evaluated our method on the real-world dataset MegaDepth and on further semantic matching datasets PF-PASCAL and PF-WILLOW, addressing concerns about generalizability. Additionally, we have provided discussion about using an alternative feature backbone, ResNet, in the sampling phase, demonstrating its potential to be agnostic to the choice of backbone. We sincerely hope that the reviewer will recognize the depth of our investigation and kindly invite the reviewer to contribute to our extended discussion.

---

### Author Response · Authors · 2023-11-23

We have updated the discussions in the manuscript to clarify points raised by the reviewers, with the revised text highlighted in red. The main changes include:

1. Updating Table 5 in the main paper to include the relationship between multiple hypotheses, matching accuracy, and time complexity.
2. Adding a discussion about the matching prior in “C.3 The Effectiveness of Generative Matching Prior” in the Appendix.
3. Including a comparison with diffusion-based dense prediction models to highlight the effectiveness of our conditioning method in “C.4 Comparison with Diffusion-Based Dense Prediction Models” in the Appendix.
4. Adding an evaluation on MegaDepth in “D.3 MegaDepth” in the Appendix to demonstrate generalizability of our model.
5. Updating minor details for clearer presentation.

We are currently conducting several experiments that we have discussed, and plan to incorporate them into the finalized version of our paper. Thank you.

---

### Meta-Review · Area_Chair_qLT5 · 2023-12-07

**Metareview:**

This work proposes a novel diffusion based network for dense pixel matching across images. All the reviewers recommend accepting the work. Reviewers appreciated the well-written paper, technical novelty and good results. Some reviewers raised several clarification questions several of which are addressed in the author responses. The reviewers did raise some valuable concerns that should be addressed in the final camera-ready version of the paper, which include adding the relevant rebuttal discussions and revisions in the main paper. The authors are encouraged to make the necessary changes to the best of their ability.

**Justification For Why Not Higher Score:**

Some minor concerns remain such as novelty and more reasoning for the design choices.

**Justification For Why Not Lower Score:**

All reviewers gave high scores of all 8s and appreciated the well-written paper.

---

### Decision · Program_Chairs · 2024-01-16

Accept (oral)